# Forward-looking insights in laser-generated ultra-intense $\gamma$-ray and neutron sources for nuclear application and science

M. M. Günther [1✉], O. N. Rosmej[1,2,3], P. Tavana[2], M. Gyrdymov[2], A. Skobliakov[4], A. Kantsyrev[4], S. Zähter [1,2], N. G. Borisenko[5], A. Pukhov [6] & N. E. Andreev[7,8]

Ultra-intense MeV photon and neutron beams are indispensable tools in many research fields such as nuclear, atomic and material science as well as in medical and biophysical applications. For applications in laboratory nuclear astrophysics, neutron fluxes in excess of $10^{21}$ n/($cm^2$ s) are required. Such ultra-high fluxes are unattainable with existing conventional reactor- and accelerator-based facilities. Currently discussed concepts for generating high-flux neutron beams are based on ultra-high power multi-petawatt lasers operating around $10^{23}$ W/$cm^2$ intensities. Here, we present an efficient concept for generating $\gamma$ and neutron beams based on enhanced production of direct laser-accelerated electrons in relativistic laser interactions with a long-scale near critical density plasma at $10^{19}$ W/$cm^2$ intensity. Experimental insights in the laser-driven generation of ultra-intense, well-directed multi-MeV beams of photons more than $10^{12}$ ph/sr and an ultra-high intense neutron source with greater than $6 \times 10^{10}$ neutrons per shot are presented. More than 1.4% laser-to-gamma conversion efficiency above 10 MeV and 0.05% laser-to-neutron conversion efficiency were recorded, already at moderate relativistic laser intensities and ps pulse duration. This approach promises a strong boost of the diagnostic potential of existing kJ PW laser systems used for Inertial Confinement Fusion (ICF) research.

[1] GSI-Helmholtzzentrum für Schwerionenforschung GmbH, Planckstraße 1, 64291 Darmstadt, Germany. [2] Goethe-Universität Frankfurt am Main, Max-von-Laue-Str.1, 60438 Frankfurt am Main, Germany. [3] Helmholtz Forschungsakademie Hessen für FAIR (HFHF), Campus Frankfurt am Main, Darmstadt, Germany. [4] Institute for Theoretical and Experimental Physics named by A.I. Alikhanov of NRC<<Kurchatov Institute>>, B. Cheremuschkinskaya 25, 117218 Moscow, Russia. [5] P. N. Lebedev Physical Institute, RAS, Leninsky Prospekt 53, 119991 Moscow, Russia. [6] Heinrich-Heine-Universität Düsseldorf, Universitätsstraße 1, Gebäude 25.32 Etage 01, 40225 Düsseldorf, Germany. [7] Joint Institute for High Temperatures, RAS, Izhorskaya st. 13, Bldg. 2, 125412 Moscow, Russia. [8] Moscow Institute of Physics and Technology (State University), Institutskiy Pereulok 9, 141700 Dolgoprudny Moscow region, Russia. ✉email: m.guenther@gsi.de

Ultra-intense $\gamma$ and neutron beams are of interest in laboratory astrophysics[1–3], high energy density plasma physics[4–9], and applications e.g. in biology, medicine, and material science[10–16].

High-brilliant $\gamma$ beams open possibilities for selective excitation of atomic nucleus states for investigation of photo-disintegration reactions in nuclear astrophysics[17] as well as efficient production of radioisotopes for medical diagnostics and radio-oncological therapeutics, such as scandium, copper, and platinum isotopes[14,15,18–20]. Also, studies of nuclear resonance fluorescence (NRF) are attractive for the development of isotope selective radiographic imaging techniques[21].

Neutron sources are indispensable tools for approving astrophysical nucleosynthesis theories, where the production of bulk heavy nuclei beyond iron (Fe) is attributed to the slow (s) and rapid (r) neutron capture processes. Nuclear reaction cross sections for nuclear structure physics studies can be accessed by means of conventional neutron sources based on electron accelerators[22], nuclear fission reactors like high-flux neutron reactor of the Institut Laue-Langevin (ILL)[23], and spallation sources[24]. The ILL reactor ($10^{15}$ n/(cm$^2$ s))[23] and the upcoming European Spallation Source (ESS) with $10^{16}$ n/(cm$^2$ s)[25] are examples for the highest peak-fluxes of the existing neutron sources. For most applications, especially in nuclear astrophysics, fast neutrons have to be moderated to the epithermal neutron energy range which decreases the effective neutron flux on target. For nuclear astrophysical studies the so-called Time-Of-Flight (TOF) facilities are preferable to reach suitable conditions for neutron energy distribution. For example, the LANSCE facility of the Los Alamos National Laboratory in the USA provides neutron peak-fluxes of $10^{15}$ n/(cm$^2$ s) as well as an integrated neutron flux of $10^5$ n/(cm$^2$s) in the nuclear astrophysical relevant neutron energy range between 10–100 keV[26]. Currently, quasi-stellar beam conditions for nuclear astrophysical investigations are available at the Frankfurt Neutron Source of the Stern Gerlach Zentrum (FRANZ) with an integrated neutron flux of $6 \times 10^6$ n/(cm$^2$ s) in the energy range of 10–100 keV on target[1,27].

Investigation of the r-process for nucleosynthesis of heavy isotopes takes place along the chain of short-lived isotopes where the neutron capture time is shorter than the time of the $\beta^-$–decay[1,28]. For r-process studies, nuclear quantities such as $\beta$-decay properties, masses, and neutron capture rates for nuclei far from the so-called valley of stability are of key importance and are still an experimental challenge[29]. One example for a suitable initial isotope for the investigation of multi-neutron capture events in the r-process chain, but also in the s-process scheme is the stable $^{96}$Zr isotope[30,31]. Chen et al.[31] have shown in simulations that for multiple neutron capture processes, a neutron peak-flux of $10^{24}$ n/(cm$^2$ s) is required to ensure the neutron capture time ($\approx 1$ s) shorter than the $^{97-102}$Zr isotope half-lives. The neutron capture time is determined by the product of the neutron peak-flux and the capture cross section. Therefore, especially for laboratory nuclear astrophysical studies the development of neutron sources with high neutron-fluxes and neutron energies from tens of keV to several hundreds of keV are of great importance. Recent theoretical studies of the neutron capture cascade using laser-driven neutron sources have shown the feasibility for neutron capture nucleosynthesis in the laboratory[32].

An alternative way for the generation of high-fluence and high-flux $\gamma$ and neutron beams in compact facilities is based on high-power lasers in relativistic laser-plasma interactions. Upcoming high-brilliance and high-intensity $\gamma$ beam facilities like the Variable-Energy Gamma Ray system (VEGA) at ELI-NP will produce tunable and narrow $\gamma$ beams via Compton backscattering of 100 TW laser pulses on relativistic electrons with energies above 700 MeV[12]. The planned upgrade of the high intense $\gamma$-ray

source, HI$\gamma$S-2, will provide up to 20 MeV $\gamma$ beams via Compton backscattering in a conventional accelerator-based free electron laser (FEL) structure[33]. Also in construction is the Gamma Factory facility at CERN that provides $\gamma$ beams with energies of several hundreds of MeV generated via the excitation of highly ionized relativistic heavy ions by lasers[13]. Such facilities allow for a renaissance in photo-nuclear research field. A review to a prospective possible pathway to reach peak neutron-fluxes of $10^{21}$–$10^{24}$ n/(cm$^2$ s) applicable for investigating nucleosynthesis in the laboratory is reported using ultra-high power (multi-PW) lasers with intensities up to around $10^{23}$ W/cm$^2$[31].

In general, two principal laser-plasma based methods are being pursued to generate $\gamma$ beams in the MeV regime. On the one hand, we consider the electron acceleration via the laser-driven plasma wake field based on the relativistic laser pulse interaction with a gas target (under dense plasma) at several tens of femtosecond pulse duration[34]. The interaction of the GeV quasi mono-energetic electrons with the contra-propagating laser pulse[35,36] leads to the generation of narrow band beams of Compton backscattered photons[37] with energies at $\gamma_e^2$ of the laser photon energy, where $\gamma_e = \frac{1}{\sqrt{1-(\frac{v_e}{c})^2}}$ is the electron Lorentz factor. On the other hand, by interaction of relativistic laser pulses with high-$Z$ solid targets, electrons can be ponderomotively accelerated in the low-dense pre-plasma region in front of the target[38,39]. Relativistic electrons are then decelerated in a high-$Z$ sample producing continuous MeV-bremsstrahlung radiation[40–42]. Concepts for producing multi-MeV highly brilliant photon beams from MeV-betatron emission of trapped GeV-electrons were theoretically discussed for the next generation of ultra-high intense lasers[43–46]. Such single laser schemes allow to investigate quantum electro-dynamic (QED) effects and QED plasmas[45,46]. Additionally, simulation studies considering a counter-propagating multi-PW laser scheme for two-side irradiation have shown promising results in the production of enhanced electron-positron plasma density for strong-field QED investigations[47].

Currently, besides neutron generation in inertial fusion processes[8,9], there are several other scenarios for the laser-based production of neutrons via nuclear reactions: Accelerated deuterons interact with heavy or light isotopes resulting in compound nucleus excitations or direct reactions. In the latter case, neutrons with energies of the half of the deuteron energies are produced[48]. The highest reaction cross sections are achieved at deuteron energies above 10 MeV. The neutron yield is peaked in a forward direction and the neutron beam has a sub-nanosecond pulse duration. For an efficient laser-driven acceleration of deuterons ultra-high-contrast and ultra-high-intense PW-class laser systems are demanded[48,49]. In the proton-induced reactions, a compound nucleus is excited. The interaction of protons with light isotopes results in a flat neutron spectrum. This is different from the interaction of protons with heavy isotopes, where the neutron spectrum peaks at low energies. The realization of a laser-driven spallation neutron source needs several tens of MeV proton energies. State-of-the-art research on these neutron production concepts are presented in[48,50,51].

In the case of photo-nuclear reactions, compound nucleus excitations take place. The neutron spectrum can be described by a Maxwellian-like distribution function with a mean kinetic energy up to several MeV and an isotropic angular distribution. Depending on the neutron generation process, the neutron source properties differ from each other in the energy distribution, the directionality and the pulse duration. Compared to the proton/deuteron induced neutron generation processes, the pulse duration of the gamma-driven neutrons is much shorter[52,53] due to the relativistic feature of the electron beam generating $\gamma$-rays. Nowadays, the high-yield laser-driven $\gamma$ and neutron-beam

production schemes are based on applications of high-intensity multi-petawatt-class laser systems, where the current state-of-the-art research is discussed in[52,53].

In this work, we present a forward-looking experimental scheme for generating ultra-intense γ and neutron beams at moderate relativistic laser intensities with high robustness and set record values of boosted γ and neutron fluences. We make feasible an effective laser-driven MeV-gamma and neutron production scheme via the laser pulse interaction with sub-millimeter thick aerogel polymer foam target systems. Laser energy-to-gamma/neutron conversion efficiencies of ≈2%/0.05% are reached, already at moderate relativistic laser intensities. For example, it allows the approach to direct nuclear astrophysical investigations in the laboratory. In addition, it promises a high diagnostic potential for existing kJ PW laser systems used for inertial confinement nuclear fusion research.

## Results

**Experiment**. Laser-driven electron beams are excellent tools for the generation of ultra-short and bright sources of particles and radiation. Experimental results on enhanced laser-driven beam generation in the multi-MeV energy range in interaction of sub-ps laser pulses of relativistic intensities with sub-mm thick low-density polymer foams promise a strong increase of the diagnostic potential of high-energy sub-PW and kJ PW-class laser systems[54–56].

In the experiment, a high energy sub-picosecond laser pulse of relativistic intensity interacts with a pre-ionized polymer foam of near-critical electron density (NCD), where super-ponderomotive electrons are produced via the direct laser acceleration (DLA)-process in the presence of strong quasi-static electric and magnetic fields[57,58]. A radial electrostatic field is created by ponderomotive expulsion of background plasma electrons caused by a relativistic laser pulse. At the same time, the current of accelerated electrons generates an azimuthal magnetic field[58]. A relativistic electron trapped in the channel experiences transverse betatron oscillations and gains the energy efficiently from the laser pulse when the frequency of the betatron oscillations becomes in resonance with the Doppler-shifted laser frequency[58].

The DLA works efficiently in NCD plasmas and with a picoseconds laser pulse duration. Different from the laser wake field acceleration (LWFA), the DLA does not generate electrons at very high energies, rather, it produces ample amounts of electrons with Maxwell-Boltzmann-like distributions carrying $\mu C$ of charge[57,59,60]. The interaction of a high number DLA-accelerated electrons with high-Z materials causes an ultra-high fluence of MeV-radiation that can trigger nuclear reactions resulting into neutron production.

The effective temperature of the DLA accelerated electrons exceeds more than one order of magnitude the ponderomotive potential and their energies extend up to 100 MeV, already at moderate relativistic laser intensities[55,56]. The reason for this behavior is a long acceleration path in a NCD plasma ensured by pre-ionized sub-mm thick foams and a relatively long sub-ps laser pulse duration. This differs from the case of the so-called multilayer targets where a low-density ("near-critical") layer of a few-micron thickness is added on the illuminated side of a thin, high-density layer. Such kind of a target was used in the last decade for proton acceleration at ultra-relativistic intensities and sub 100-fs pulses[61–63]. In[62], the generation of hot electrons from a double layer target irradiated by a laser pulse with an intensity of $2 \times 10^{20}$ W/cm$^2$ was reported. The target consisted of a several micrometers thin NCD plasma slab (0.5 $n_{cr}$) and an over-dense foil. The measured effective electron temperature was 8 MeV, which is only twice higher than the ponderomotive potential.

The present experiment was performed at the high-energy laser facility PHELIX[64]. We investigated the electron acceleration with the focus on γ and neutron production in the interaction of DLA electrons with high-Z materials using two different peak laser intensities (~$10^{19}$ W/cm$^2$ and ~$10^{21}$ W/cm$^2$). These different intensity regimes were achieved by different focusing systems[56]. Polymer aerogel-foams with a mean density of 2 mg/cm$^3$ and sub-millimeter thickness[65] were used for the production of a NCD-plasma via the mechanism of super-sonic ionization[55,66] by sending a well-controlled nanosecond-pulse before the main relativistic pulse. A fully ionized plasma corresponds to $0.64 \times 10^{21}$ cm$^{-3}$ electron density or 0.64 $n_{cr}$ ($n_{cr} = 10^{21}$ cm$^{-3}$). Measurements of the plasma density inside the ringholder after ionization by a super-sonic wave is a big challenge. At the same time, Particle-In-Cell(PIC) simulations performed for a step-like density profile with $n_{cr}$ and 0.5 $n_{cr}$[59,60] as well as for a partially ramped density profile in order to account for plasma expansion toward the main laser pulse[55] showed a very similar overall behavior of the energy and angular distributions of super-ponderomotive electrons.

Compared to NCD-plasmas generated by laser irradiation of conventional foils, the DLA path in 300–400 μm thick foams is strongly enhanced, which results in an increased propagation length of the laser main pulse through the NCD-plasma and allows to produce a high charge of electrons with energies far above the ponderomotive potential[55]. A schematic of the experimental setup is presented in Fig. 1. Here, the energy distribution of the DLA-electrons was measured by means of magnet spectrometers. A steel cylinder was used to map the angular distribution of the electron beam. The high-current, well-directed, super-ponderomotive electrons produced in the NCD plasma pass through a high-Z converter generating bremsstrahlung. γ-rays together with some fraction of relativistic electrons that escape the converter, propagated 23 cm in vacuum and triggered nuclear reactions in the activation detectors consisting of gold (Au), tantalum (Ta), indium (In), and chromium (Cr).

For further discussion, we introduce the convention for the designations of the four principal laser-target setups used in the experiment, which will be applied in the following: setup 1a ($10^{19}$ W/cm$^2$ onto foam + thin foil), setup 1b ($10^{19}$ W/cm$^2$ onto

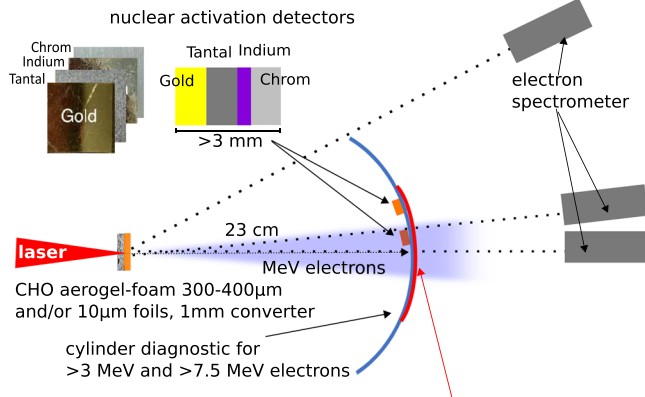

**Fig. 1 Experimental set-up.** Top view of the diagnostic set-ups used for irradiation of aerogel-foams (CHO) at $10^{19}$ W/cm$^2$ (20 J in focal spot) laser intensity and conventional foils at ultra-relativistic intensity of ~$10^{21}$ W/cm$^2$ (40 J in focal spot). An imaging plate (IP) behind 6 mm thick steel cylinder was used to measure an angular distribution of electrons with energies above 7.5 MeV. Nuclear activation plates consisting of different materials were placed at the front of cylinder at 5° and 15° to the laser axis. Three 0.99 T magnetic spectrometer measured the energy distribution of accelerated electrons at 0°, 15° and 45° to the laser axis.

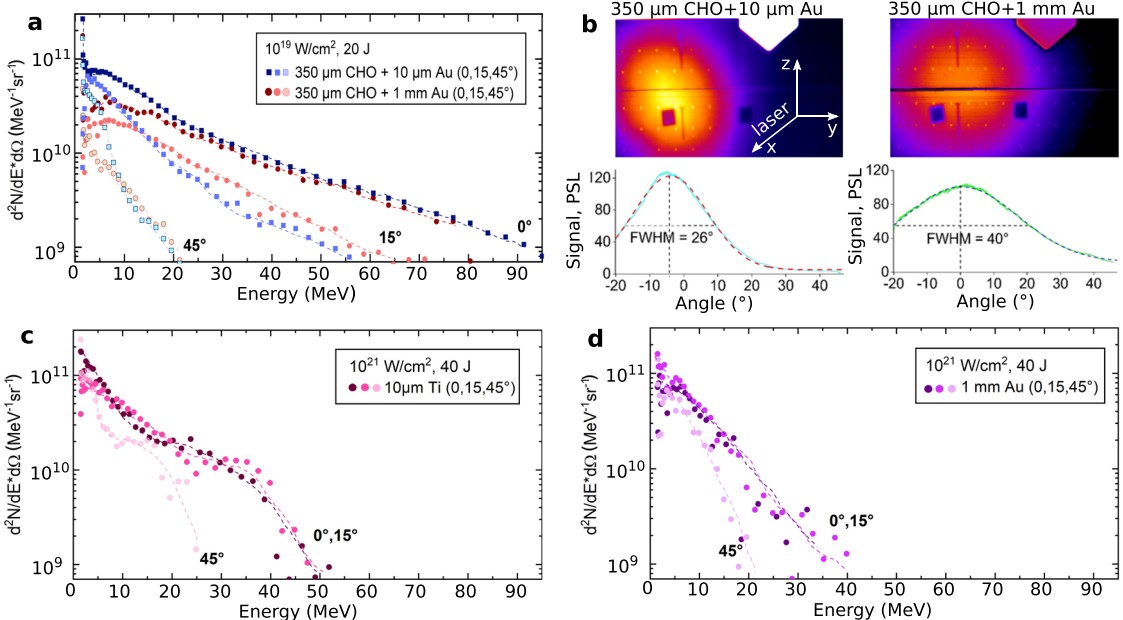

**Fig. 2 Electron spectra.** Electron spectra in values of the fluence per electron energy as function of electron energy registered at three different angles (0°, 15°, 45°) to the laser direction: **a** Spectra measured in interaction of a laser pulse with $10^{19}$ W/cm² intensity and 20 J in the FWHM of the focal spot with aerogel foam (CHO) target stacked together with a gold (Au) foil of 10 μm thickness (setup 1a, blue) and with 1 mm thick gold converter (setup 1b, red). **b** Result of the cylinder stack diagnostic for the angular distribution of super-ponderomotive electrons with energies above 7.5 MeV measured in a single shot onto aerogel + thin foil (left) and aerogel + 1 mm-thick converter (right). **c** Spectra measured in interaction of a laser pulse with $10^{21}$ W/cm² and 40 J in FWHM of the focal spot with 10 μm titanium (Ti) foil and **d** with 1 mm-thick gold converter.

foam + 1 mm thick foil), setup 2a ($10^{21}$ W/cm² onto thin foil), setup 2b ($10^{21}$ W/cm² onto 1 mm thick foil).

**Electron spectra.** In the experiment, spectra of super-ponderomotive electrons were measured by means of magnetic spectrometers at three different angles to the laser axis (see Fig. 1). In the case of the setup 1a and 1b, we observe a strong dependence of the electron energies on the angle between the electron beam propagation direction and the laser direction, which is presented in Fig. 2a. The experimental graphs are shown for the electron spectra measured at 0°, 15° and 45° to the laser axis for setup 1a (blue) and setup 1b (red). The main fraction of relativistic electrons is accelerated along the laser axis (0°) up to energies of 100 MeV (Fig. 2a). The difference in the spectra between the setup 1a and 1b is caused by propagation of electrons through 1 mm thick Au-convertor in the 1b-case. This thickness corresponds to the mean free path of electrons with $E \le 8$ MeV.

The electron energy distribution was approximated by a Maxwellian-like distribution function with one or two effective temperatures $T_{hot}$. In the case of setup 1a, the electron spectrum measured at 0° was described by an exponential function with $T_{hot,1} = (12.1 \pm 1.4)$ MeV and $T_{hot,2} = (26.5 \pm 3.4)$ MeV, both exceeding the ponderomotive potential at $10^{19}$ W/cm² more than one order of magnitude. With increasing the observation angle, the temperature and the number of accelerated electrons drop down to $T_{hot,1} = (8.1 \pm 1.2)$ MeV ($T_{hot,2} = (15.8 \pm 1.7)$ MeV) at 15° and further to $T_{hot,1} = (1.9 \pm 0.1)$ MeV ($T_{hot,2} = (7.3 \pm 1.2)$ MeV) at 45°. Electron spectra measured in the case of setup 1b show the same tendency. The difference between 1a and 1b is seen only below 15 MeV for electrons that undergo significant energy loss in the 1 mm thick Au-converter (see Fig. 2a).

In direct shots on thin metallic foils at ultra-relativistic laser intensity of $10^{21}$ W/cm² (setup 2a), the electron energy distribution was approximated with $T_{hot} = (9.9 \pm 2.2)$ MeV (Fig. 2c) for measurements at 0° and 15° to the laser axis and the maximum of

the detected electron energy reached 50 MeV, which is twice lower than in the case 1a and 1b. In shots onto 1 mm thick converter (setup 2b), the effective temperature and the maximum of the detected energy of escaping electrons are 4 MeV and 40 MeV correspondingly (Fig. 2d). The dependence of the electron energy distribution from angle to the laser axis is observed only at 45°, which indicates a rather divergent electron beam. In shots with $10^{21}$ W/cm² laser intensity, 100 μm entrance slit was used (see Methods, Electron diagnostics). This resulted into comparable level of the electron signal and the background caused by the bremsstrahlung radiation from the Au-converter and, as a consequence, rather noisy electron spectra.

Measurements using cylinder diagnostic (see Methods) showed that the electron beam generated via DLA process in NCD plasma is well-collimated. Fig. 2b presents results of 120°- cylinder diagnostic for setups 1a and 1b. The Imaging Plate (IP) signals are caused by electrons with energies above 7.5 MeV, which are capable to trigger nuclear reactions in activation samples. The measurements show that in the first case, the relativistic DLA electrons with energies above 7.5 MeV propagate within a half angle of 13° to the laser axis (0.16 sr solid angle), while in the setup 1b within 20° (0.38 sr) due to electron scattering in 1 mm thick foil. From shot-to-shot deviation of the electron beam position from the laser axis was not larger than 5°. This result is in contrast to the stochastic beam pointing reported by Willingale et al.[54].

**MeV γ beam.** For the characterization of the bremsstrahlung spectrum generated by super ponderomotive electrons in high-$Z$ samples, we used a nuclear activation-based diagnostic, which is sensitive to more than 7.5 MeV photon energies, similar to that described in[67]. In the present work, the samples composed of elementary Au, Ta, In and Cr were stacked together and placed at a distance of 23 cm in forward direction at 5° and 15° to the laser axis (Fig. 1).

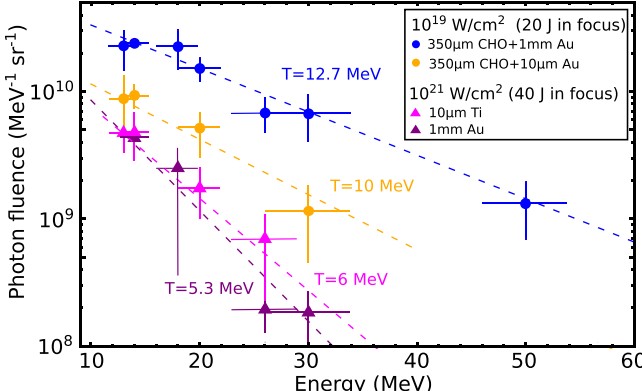

**Fig. 3 MeV-bremsstrahlung.** MeV-bremsstrahlung fluence in dependence on energies from different laser-target setups described in the legend, where used combined aerogel (CHO) with two different thick gold (Au) targets (data as blue and yellow dots) as well as thin and thick metal targets, only (data as magenta and purple triangles). The error bars represent the accuracy of reaction yield and the width of nuclear resonance used for spectral reconstruction (see Methods). Dashed lines are thermal exponential fits on the experimental data.

After each laser shot, the nuclear-activated samples were analyzed by a low-background spectroscopy using 60% high purity germanium (HPGe) detector systems to identify the reaction channels leading to the isotopes of interest and to determine reaction yields of the activated isotopes in every sample (see below, methods nuclear diagnostics). The high-energy bremsstrahlung spectrum was evaluated according to the analysis process described in Methods.

Figure 3 shows the energy-dependent bremsstrahlung fluences detected at 5° to the laser axis for the different laser intensities and target systems (setup: 1a, 1b, 2a, and 2b). In the case of setup 1b, the photon spectrum follows an exponential dependence on energy with an effective temperature of $(12.7 \pm 2.1)$ MeV. The measured number of photons reaches $(4.9 \pm 2.1) \times 10^9$ per $1.5 \times 1.5$ cm$^2$ area of the activation sample stack or $(1.2 \pm 0.5)$ $10^{12}$ ph/sr. We would like to point out, that only in interaction of $10^{19}$ W/cm$^2$ laser pulse with a foam layer stacked together with 1mm thick Au-converter (1b), photo-neutron disintegrations up to $^{197}$Au$(\gamma,5n)^{192}$Au with maximum of the reaction cross section at 50 MeV (see below) were observed (blue dots at 50 MeV, Fig. 3).

In the case of direct irradiation of a high-$Z$ converter with ultra-relativistic laser intensity (setup 2b), a 10 times lower fluence of $(1.2 \pm 0.5)10^{11}$ ph/sr with a twice lower effective temperature of $(5.3 \pm 2.0)$ MeV was recorded compared to 1b (see Fig. 3). Here, only $(\gamma,3n)$-reactions in Au and Ta with maximum of the reaction cross section at ~30 MeV were registered.

The difference in the photon fluences and effective temperatures in 1b- and 2b- cases can be explained by higher electron energy and directionality of the relativistic electron beam generated in the interaction of $10^{19}$ W/cm$^2$ (20 J) laser pulse with pre-ionized sub-mm thick foam targets compared to shots with an ultra-relativistic intensity of $10^{21}$ W/cm$^2$ (40 J) onto Au-converter (see Fig. 2a and d).

The bremsstrahlung spectra measured by means of the nuclear activation method in setup 1b and 2b consist of a primary gamma-beam with energies above 10 MeV (threshold of photo-nuclear reactions), generated by laser-accelerated electrons in the Au-converter, and a secondary $\gamma$-beam, produced by above 10 MeV electrons that emerge the converter (see Fig. 1, experimental setup) and hit the activation stack. The acceptance angle

of the activation sample stack is of 4.3 msr, only. This is a small part of solid angles covered by the relativistic electron and $\gamma$ beams. In the case of 1b, the divergence angle of electrons was defined experimentally (Fig. 2b), the divergence angle of the $\gamma$ beam with energies above 10 MeV generated in the converter was unknown.

In order to solve this problem, the GEANT4 Monte Carlo code[68] was used to simulate the relativistic electron transport through a 1 mm thick converter (setup 1b) and the generation of the primary $\gamma$ beam. In GEANT4, the following physical processes were involved in simulations of the relativistic electron beam interaction with high-$Z$ converter and activation plates: collisional and radiative energy loss, $e^-e^+$ pair production, gamma driven nuclear reactions in the region of Giant Dipole Resonance (GDR). In simulations, ShieldingLEND[69] physics list was applied which represents Low Energy Nuclear Data (LEND) with a set of low energy nuclear interaction processes. Electro-nuclear reactions in GEANT4 are handled by the classes G4ElectroNuclearProcess and G4PositronNuclearProcess[69]. The experimental geometry together with the measured energy and angular distributions of the DLA-electrons in the case of setup 1a were used as input parameters. A converter plate was considered as cold solid matter, since the laser energy was fully absorbed in the foam[56] and the laser did not hit the converter. The benchmarking was done by comparing measured and simulated isotope yields (see below, methods nuclear diagnostics). Simulations resulted in a half divergence angle of 17° (0.27 sr) for electrons above 10 MeV and 10.5° (0.11 sr) for $\gamma$ of the same energy (see Fig. 4). This means that in the case of setup 1b, 1.6% of electrons and 4% of primary gammas generated in 1 mm Au converter interact with the activation samples and trigger nuclear reactions. In addition, in the sample placed at 15°, a more than one order of magnitude lower activity was measured, which also indicates a highly directed MeV-bremsstrahlung beam. The differences between the measured (0.38 sr, Fig. 2b) and the simulated (0.27 sr) electron divergence cones can be explained by a strong increase of the IP-response produced by electrons with energies below 1 MeV[70] compared to those above 1 MeV. The beam of DLA-electrons passes a 1 mm thick Au-converter and produces the MeV-bremsstrahlung radiation with 0.11 sr divergence angle. A high fraction of the relativistic electrons escapes the converter (Fig. 2, setup case 1b) with 0.27 sr divergence, propagate 23 cm in vacuum and produce additional gamma-radiation in the activation samples. According to GEANT4 simulations, 65% of the total gamma-fluence originates in the converter and 35% in the activation stack. Under consideration, that $1.2 \times 10^{12}$ ph/sr are produced in two different ways described above, we obtain the total photon number with energies greater than 10 MeV: $1.2 \times 10^{12}$ ph/sr $\times (0.65 \times 0.11$ sr $+ 0.35 \times 0.27$ sr$) = 2 \times 10^{11}$. Taking into account the effective temperature of $(12.7 \pm 2.1)$ MeV and laser energy of 20 J, we end up with an ultra-high conversion efficiency of the laser energy into above 10 MeV gammas of $(1.4 \pm 0.3)$%. Compared to the present work, a concept of a non-linear Compton back-scattering scheme driven by a PW short laser pulse at more than $5 \times 10^{21}$ W/cm$^2$ in a NCD plasma coupled with a plasma mirror was recently discussed in[71]. The described simulation studies show a broad-band spectrum that contains more than $10^{11}$ photons with energies above 10 MeV with less than 1% laser-to-$\gamma$ energy conversion efficiency.

**Accelerated protons.** In laser interactions with thin foil targets at $10^{21}$ W/cm$^2$ laser intensity (setup 2a) and with combined foam and thin foil target systems at $10^{19}$ W/cm$^2$ (setup 1a), we observed proton induced nuclear reactions in the GDR region.

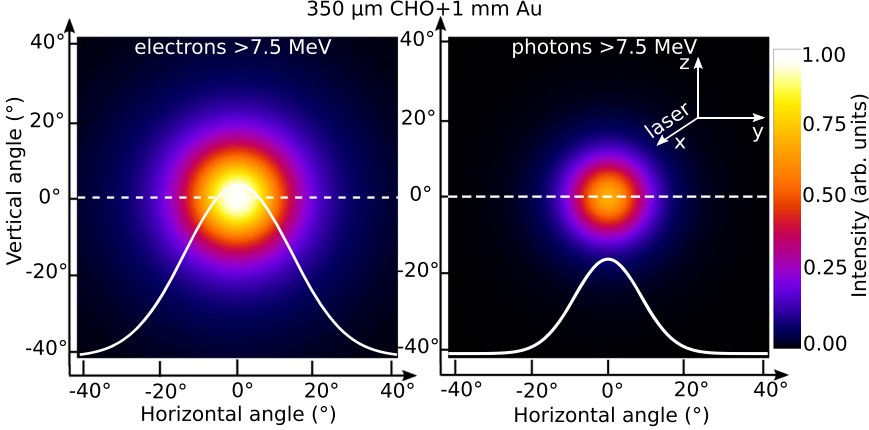

**Fig. 4 Simulated angular distribution.** GEANT4 simulations of the angular distribution of electrons and photons above 7.5 MeV after the laser pulse interaction with an aerogel-foam (CHO) target combined with a 1 mm thick converter (Au). The angular distributions are shown using the same color scale in arbitrary unit. The half divergence angle of the electron beam is 17°, where the photons are distributed within the half divergence angle of 10.5°. The white curves represent the line-out (dashed lines) of the colored distribution in the range of the horizontal angle in degree (°).

These reactions lead to compound nucleus excitations in Au and Cr activation samples and generation of mercury (Hg) and manganese (Mn) isotopes. The reaction cross sections and corresponding proton energy ranges of the GDRs are shown below in the methods section. In the case of proton-induced reactions, there are no competing photo-nuclear channels that lead to the same isotope if starting from the same natural stable mother isotopes.

In the case where the activation stack was oriented to laser with Au-sample (Cr is at rear), we observed following proton-driven nuclear reactions: $^{197}Au(p,n)^{197m}Hg$ with a maximum of the cross section at 12 MeV proton energy, $^{197}Au(p,3n)^{195m}Hg$ at 28 MeV and $^{54}Cr(p,3n)^{52}Mn$ at 37 MeV. For the opposite sequence of samples (Cr is at front), we identified the reaction channel $^{52}Cr(p,n)^{52}Mn$ at 14 MeV. The gamma spectra of the different sequenced activation stacks are shown in the methods section (see below).

The proton number within the width of each GDR was deduced from the reaction yield of the corresponding isotope by deconvolution with the reaction cross sections (see below, methods nuclear diagnostics). The proton number per sample measured in the GDR region centered at 14 MeV with the resonance width of 8 MeV and at 37 MeV with the resonance width of 20 MeV are $(1.38 \pm 0.62) \times 10^9$ and $(1.48 \pm 0.80) \times 10^9$ for the setup 1a. In the case of setup 2a, the proton number reached $(1.62 \pm 0.41) \times 10^{10}$ at 14 MeV and $(3.74 \pm 1.03) \times 10^8$ at 37 MeV proton energy. Compared to setup 2a at $10^{21}$ W/cm$^2$ and laser pulse energy of 40 J, the proton number in setup 1a at $10^{19}$ W/cm$^2$ with 20 J laser energy reaches a 4 times higher value in the upper cut-off region of proton energy. This behavior allows for enhanced nuclear reaction yields over a wide range of proton energies.

Figure 5 a shows normalized proton number in dependence on the energy obtained for setups 1a (yellow) and 2a (magenta). In the case of interaction of $10^{21}$ W/cm$^2$ laser pulse with a thin foil (setup 2a), experimental data were fitted by an exponential function with an effective temperature of 5–6 MeV. This protons spectral properties are characteristic for the target normal sheath acceleration (TNSA) mechanism[72,73]. The important feature of the proton spectrum evaluated for setup 1a is a very flat slope (Fig. 5a, yellow dashed line), which results into a high proton number at energies above 20 MeV. The dark blue curve in Fig. 5a presents the result of the 3D PIC simulation performed by means of the virtual laser plasma laboratory (VLPL) code[74] for the

experimental conditions discussed above. The slope of the simulated proton spectrum is in a good agreement with the experiment.

Another important feature of the proton beam in the case of setup 1a is its low divergence. We measured a 500 times higher $^{52}Mn$ isotope yields produced in the activation stack (Cr-front) placed at 5° to laser axis compared to those measured at 15°. In these shots, the target normal was counterclockwise rotated by 10° to laser direction (Fig. 5b). The half protons divergence angle of 6°–7° was estimated assuming a Gaussian angular distribution of the proton beam centered at the target normal and taking into account proton driven reaction yields measured at 5° and 15°.

This is in a good agreement with 3D PIC results. According to the simulations (setup 1a), the 20 J laser pulse is absorbed in the foam before it reaches the foil[56]. A large space charge of superponderomotive electrons accelerated in the NCD-plasma reaches the rear side of the foil, which stays cold. The simulations show that energetic electrons spread out along the foil rear side generating a very planar electrostatic field. This field accelerates a highly collimated ion beam. Result of simulations is presented in the diagram of the longitudinal and transversal proton momenta (Fig. 5c), which is characteristic for a highly collimated proton beam with a 5°–7° half of divergence angle.

**Neutrons.** Enhanced neutron fluences were observed in laser pulse interactions with foam target systems at moderate relativistic laser intensities. Here, in the setup 1a, neutron generation was dominated by protons, while in the setup 1b the neutrons are dominantly generated by photo-nuclear reactions.

In both cases, the generation of neutrons was investigated via a well-established indium activation method (e.g.[75]), which can be applied for fast and epithermal/slow neutrons. In the experiment, elementary indium with a purity of 99.999% in the natural isotope abundance of $^{115}In$ (95.7%) and $^{113}In$ (4.3%) was used. We focused on the decay channels of activated indium isotopes that were reached via neutron-induced reactions. In this procedure, an input of $\gamma$-driven nuclear reactions leading to the same indium isotope type was taken into account (see below, methods nuclear diagnostics).

In the study of reactions triggered by epithermal neutrons, the strong nuclear resonance with a reaction cross-section of $10^4$ barn at 1 eV for the neutron capture reaction $^{115}In(n,\gamma)^{116m}In$ was considered. Fast neutrons were investigated by the pure neutron activation channel $^{115}In(n,n')^{115m}In$ with 0.343 barn at 2.5 MeV

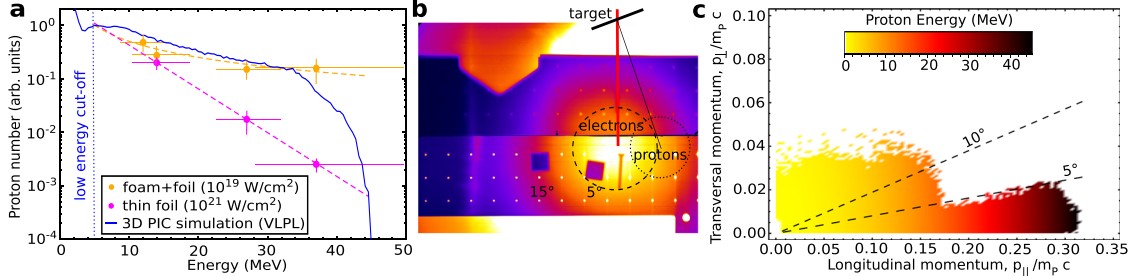

**Fig. 5 Protons. a** Proton number in arbitrary units as a function of proton energy. Yellow dots are proton number evaluated in case of laser interaction with aerogel-foams combined with thin metal foils at $10^{19}$ W/cm² intensity (setup 1a), while the magenta dots describe proton number obtained in laser interactions with thin metal foils at $10^{21}$ W/cm² laser intensity (setup 2a). Error bars represent the accuracy of reaction yield and the width of nuclear resonance used for spectral reconstruction. The dashed curves demonstrate the different slopes of proton spectra. PIC-simulated proton spectrum is shown as blue solid line. **b** IP signal produced by electrons above 7.5 MeV directed along the laser axis (FWHM range, black dashed circle) with positions of activation samples for detection of (p,xn)-reactions and expected direction of TNSA-accelerated proton beam (foil-target normal was +10° to laser direction). The half protons divergence angle of 6°–7° was estimated assuming a Gaussian angular distribution at the target normal and taking into account proton-induced reaction yields measured at 5° and 15° (black dotted circle). **c** Momentum diagram of protons from 3D PIC simulation (VLPL code). Transversal and longitudinal momenta are shown as normalized values, where the colors display the proton energies. Dashed lines indicate the half divergence angle ranges at 5° and 10°.

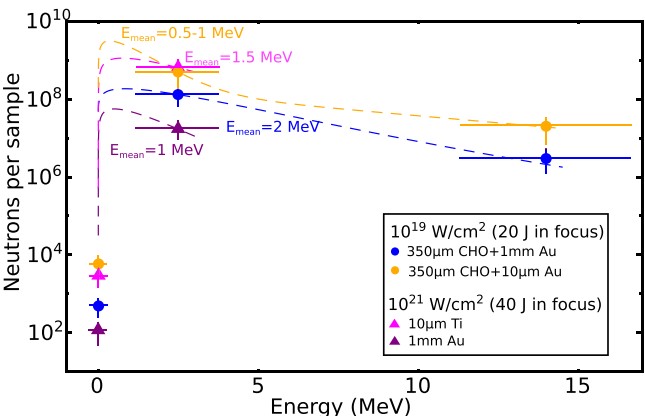

**Fig. 6 Neutrons.** Number of neutrons in different Giant Dipol Resonance (GDR) regions measured in the activation sample with 4.3 msr acceptance angle for different laser-target setups described in the legend. The error bars represent the accuracy of reaction yield and the width of nuclear resonance used for spectral reconstruction (see Methods). The mean neutron energies $E_{mean}$ are evaluated from the activation method (see methods) and used to build a Maxwellian-like energy distribution of neutrons (dashed lines).

and the activation channels $^{113}$In(n,n')$^{113m}$In with 0.2 barn at 2.5 MeV as well as the $^{113}$In(n,2n)$^{112m}$In reaction with 1.5 barn at 14 MeV initial neutron energy. The reaction channels $^{115}$In(n,2n) $^{114m}$In and $^{113}$In(n,γ)$^{114m}$In are not useful because the daughter isotope becomes the same for fast and slow neutrons.

The neutrons were produced within the activation sample placed at a distance of 23 cm to the laser interaction point, which covers a solid angle of 4.3 msr. According to GEANT4 simulations, a contribution of background neutrons to the nuclear activation of In isotopes was not higher than 30%.

In the case of setup 1b only, we observed a significant yield of $^{112m}$In and $^{114m}$In isomers produced by fast neutrons generated in γ-driven nuclear reactions triggered by a high number of above 10 MeV photons (Fig. 3, setup 1b). Such reactions lead to the production of neutrons, which are evaporated with high kinetic energy from an excited compound nucleus.

Figure 6 shows the neutron number per activation sample evaluated using the indium isotope yields after subtraction of the

γ-driven channels. The energy distribution of neutrons evaporated from excited states of compound nucleus can be described by a Maxwellian-like function[76]. The neutron number per energy width of the corresponding GDR was determined by weighting the reaction yields with the cross sections[77] for each reaction. As described in the methods, the mean neutron energy $E_{mean}$ was evaluated by an isotope relation equation using the experimentally measured reaction yields of indium isotopes produced by epithermal as well as fast neutrons starting from the same mother isotope similar it was shown in[75]. The total neutron numbers were obtained by fitting the measured neutron spectra with a Maxwellian-like distribution function $N(E/(\pi E_{mean}))^{1/2} \exp(-E/E_{mean})$ using the known $E_{mean}$, where $E$ is the neutron kinetic energy. The dashed curves in Fig. 6 present such neutron distribution functions for each scenario.

In the case of setup 1b (Fig. 6, blue dots), the enhanced neutron production centered at fast neutrons is explained by a higher effective temperature and fluence of the MeV-bremsstrahlung (see Fig. 3). In this case, we observe neutron activations in indium around 14 MeV neutron energy.

In the setup 1a and 2a, using thin foils, neutrons were produced by proton-induced nuclear reactions. Only in the case of setup 1a, we observed a high number of $^{116m}$In generated in interactions with epithermal/slow neutrons (see below, methods nuclear diagnostics) and $^{112m}$In activations at 14 MeV neutron energy. This behavior is a consequence of the flat slope of the proton spectrum, which leads to a high proton number at energies greater than 20 MeV responsible for fast neutron production. The neutron spectrum in the case of 1a can be explained by a combined Maxwellian-like distribution with an additional exponential part describing the high energy tail of the neutron spectrum[76,78,79]. Here, the process of precompound excitation in proton-induced nuclear reactions is taking into account, where the step after the nuclear reaction leads to the probability to emit a nucleon (neutron) immediately before the compound nucleus state[80–82]. Such probability increases with the proton energy. Therefore the flat slope of the proton spectrum obtained in the setup case 1a, favors the probability for precompound excitations. Neutrons emitted from the precompound state have the most energy but very few in number compared to the statistical evaporated neutrons from compound state, which means a small contribution to the neutron energy distribution[78,79]. The observed reaction yields in In fit well with GEANT4 simulations (see below, methods).

In addition to the indium isomer activation, we observed neutron capture reactions in the Au sample ($^{197}Au(n,\gamma)^{198m}Au$) induced by epithermal neutrons at 5 eV, where a strong resonance with a cross section of $2.8 \times 10^4$ barn was excited. The highest yield of $^{198m}Au$ was observed for the laser-target setup 1a, where the mean kinetic energy of neutrons generated in proton driven nuclear reactions was of 0.5–1 MeV. The stable Au isotope represents a waiting point within the nuclear astrophysical r-process heavy element nucleosynthesis. Within the experimental accuracy, the $^{198m}Au$ yield reached in the present work is in a good agreement with the theoretical study by Hill et al.[32].

Summarizing the results, we suggest an universal approach for a strongly enhanced laser-based generation of ultra-high neutron fluence triggered by gammas and/or protons using foam-foil target systems. Keeping the laser parameters constant and varying the target systems one can create two neutron production schemes:

1. In the case of aerogel-foam targets combined with 10 μm Au foils, the neutron generation is dominated by proton-induced nuclear reactions. The neutron number evaluated from the proton-driven nuclear reactions in the sample with 4.3 msr acceptance angle reaches $(6.7 \pm 0.8) \times 10^9$ per shot, while the mean neutron kinetic energy was of 0.5 – 1 MeV. Triggered by all accelerated protons propagating in the solid angle of 0.04 sr the neutron number reaches $(6.2 \pm 0.5) \times 10^{10}$ per laser shot.

2. Using aerogel-foams stacked with high-$Z$ converters (setup 1b), the neutron generation via photo-nuclear reactions in the sample is caused by high fluence electrons and $\gamma$'s with energies above 10 MeV. The evaluated neutron number is of $(3.6 \pm 0.6) \times 10^8$ per shot and a mean neutron kinetic energy of 2 MeV. The number of neutrons triggered by all initial electrons and photons above 10 MeV propagating in 0.27 sr and 0.11 sr solid angles (Fig. 6) reaches $(1.4 \pm 0.2) \times 10^{10}$ per laser shot.

The high nuclear reaction yield in the case of aerogel targets and the corresponding total fluence and temperature of $\gamma$ photons as well as mean kinetic energy of neutrons is realized by the strong enhancement of the number and the energy of super-ponderomotive electrons compared to shots onto foils at ultra-relativistic intensity.

In Table 1, the results of the present work are shown in comparison with previous experiments on laser-driven neutron sources, where record values were reported in two independent neutron production schemes described by Roth et al. and Jung et al.[48,49], Kleinschmidt et al.[51], and by Pomerantz et al.[52]. In the present work, using the combination of low density foams with thin or thick metallic foils, two neutron generation schemes were realized: one via pure $\gamma$ induced nuclear reactions and another one, via proton-dominated nuclear reactions, which results in a high neutron fluence.

In setup 1b, a high current beam of super-ponderomotive electrons boosts the production of MeV-bremsstrahlung radiation by propagation through a high-$Z$ converter. In this scheme, a record conversion efficiency of laser energy into energy of more than 10 MeV $\gamma$-rays of $(1.4 \pm 0.3)\%$ was achieved. The measured $1.4 \times 10^{10}$ neutrons with 2 MeV mean energy correspond to 0.02% laser-to-neutron conversion efficiency. In[52], super-ponderomotive electron beams were used to trigger neutron production in Cu-converter. The relativistic electrons were generated at $5 \times 10^{20}$ W/cm$^2$ laser intensity and 90 J laser energy on target in interaction with underdense CH-plasma. For comparison with the presented results we assume a similar laser energy in the focal spot of 20–30 J. The number of registered neutrons and the corresponding conversion efficiency were more than 10 times lower than in our case (see Table 1).

The more efficient neutron production scheme can be realized via proton and deuteron-induced nuclear reactions. In our experiment, using foam + thin foil (setup 1a), we observed enhance generation of protons in the GDR region above 20 MeV and measured $6.2 \times 10^{10}$ neutrons with 0.5–1 MeV mean energy. The laser-to-neutron conversion efficiency of 0.05% exceeds results reported in[48,49] and is close to those obtained at the PHELIX-laser by Kleinschmidt et al.[51] using deuteron induced nuclear reactions. Note that in all experiments, results were obtained at higher laser energies and at 10–50 times higher laser intensities than in our case.

An additional advantage of our approach realized via proton beams, is the generation of a high number of neutrons with moderate kinetic energies, which can be applied in experiments on laboratory astrophysics. Confirmation of this is the observation of the high yield of $^{198m}Au$ produced in the neutron capture process.

In summary, the interaction of a sub-ps laser pulse of moderate relativistic intensity with sub-mm long polymer foams combined with thin and thick high-$Z$ foils results into the highest number of generated neutrons per Joule laser energy compared to the schemes presented in Table 1. This concept promises a strong boost of neutron production on kJ PW-class laser systems.

**Table 1 Conversion efficiencies.** Results from presented work in case of $\gamma$-neutron production and proton-driven neutron generation are shown in comparison with previous experiments, where records of neutron generation via laser accelerated deuterons interacting with beryllium catcher[48,49,51] as well as via the electron acceleration to bremsstrahlung to neutron scheme[52] were reported.

| | p-driven(this work) | γ-driven(this work) | ion-driven[48,49] | ion-driven[51] | γ-driven[52] |
|---|---|---|---|---|---|
| laser intensity $I_{laser}$ | ~$10^{19} \frac{W}{cm^2}$ | ~$10^{19} \frac{W}{cm^2}$ | $5 \times 10^{20} \frac{W}{cm^2}$ | $2 \times 10^{20} \frac{W}{cm^2}$ | $2 \times 10^{20} \frac{W}{cm^2}$ |
| focused energy $E_{laser}$ | 20 J | 20 J | 52 J | 60 J | 20 J – 30 J |
| target system | CHO foam | CHO foam | $CD_2$ foil | $CD_2$ foil | CH foil |
| neutron production | (p,xn) | (γ,xn) | (d,n) | (d,n) | (γ,xn) |
| primaries (>10 MeV) | ~$3 \times 10^{11}$, proton | $3 \times 10^{11}$, $e^-$ | >$10^{11}$, d | N/A, d | N/A, $e^-$ |
| laser-primary conversion eff. | – | 10% | 0.5% | N/A | N/A |
| laser-γ conversion eff. | – | 1.4% (>10 MeV) | – | – | N/A |
| total neutron number | $6.2 \times 10^{10}$ | $1.4 \times 10^{10}$ | $7.2 \times 10^9$ | $6.5 \times 10^{10}$ | $1.2 \times 10^9$ |
| neutron direction | isotropic | isotropic | 16% directed | 10% directed | isotropic |
| mean neutron energy | 500 keV–1 MeV | 2 MeV | >10 MeV directed 2–4 MeV isotropic | >10 MeV directed 2–4 MeV isotropic | ~1 MeV |
| laser-neutron conversion eff. | 0.05% | 0.02% | 0.01% | 0.07% | 0.001% |
| neutrons per laser energy | $3.1 \times 10^9$ n/J | $7 \times 10^8$ n/J | $1.3 \times 10^8$ n/J | $1.07 \times 10^9$ n/J | $6 \times 10^7$ n/J |

## Discussion

In this work, we presented a prospective way for producing well-directed high intense multi-MeV $\gamma$ beams and ultra-high fluence neutron sources at moderate relativistic laser intensities of $10^{19}$ W/cm$^2$ and 20 J energy within the focal spot. Our approach is based on the DLA-process in long-scale NCD plasmas, where the super-ponderomotive electrons with energies up to 100 MeV are produced. The charge carried by electrons with more than 10 MeV propagating in the 0.16 sr solid angle reaches 50–100 nC what corresponds to 10% of the conversion efficiency. This high-current well-directed electron beam is the basis for the generation of proton- and $\gamma$-sources applicable for the discussed nuclear research.

Ultra-high intense, highly directed MeV $\gamma$ beams with fluences greater than $10^{12}$ ph/sr above 10 MeV photon energy and effective temperatures of 13 MeV were detected. The conversion efficiency from laser energy into above 10 MeV-photons reached $(1.4 \pm 0.3)$%. This provides a basis for the preparation of a narrow band $\gamma$ beam, suitable for nuclear photonics applications[10,83].

The presented experimental technique allows to choose the generated neutron behavior by varying the target system. Depending on the application needs, the laser-driven neutron source of ultra-high fast neutrons can be provided via photo-nuclear reactions with a laser energy conversion efficiency of 0.02%. Higher laser to neutron conversion efficiency of 0.05% was achieved via proton-induced neutron generation already at moderate relativistic laser intensities and ps pulse duration, where the averaged neutron energies are in a suitable range for nuclear physics applications without moderation processes. The experimental top yields, realized in the present work at $10^{19}$ W/cm$^2$ laser intensities and 20 J laser energy, reached more than $10^{10}$ neutrons per laser shot in the photon-driven as well as greater than $6 \times 10^{10}$ per shot in the proton-driven scheme. The shape of the neutron spectra evaluated in this work for (p,n)-reactions is very similar to those simulated in[31] and discussed for investigation of multiple neutron capture processes in Zr.

GEANT4 Monte Carlo simulations were performed to optimize the neutron production for PHELIX parameters ($10^{19}$ W/cm$^2$, $E_{\mathrm{FWHM}}$= 20 J, setup 1b). After placing the Au converter direct to the foam rear-side and increasing the converter thickness up to 5–7 mm, a record neutron fluence of $2 \times 10^{11}$ cm$^{-2}$ and $10^{22}$ cm$^{-2}$ s$^{-1}$ flux estimated for ~20 ps neutron pulse duration was achieved. From Particle-In-Cell (PIC) simulation studies we expect a more than one order of magnitude higher relativistic electron number with energies above 7.5 MeV by increasing the laser energy from 20 J to 200 J. This results in a corresponding enhancement of the neutron fluence and flux.

Current developments in the laser technology toward higher laser pulse energies and higher repetition rates (one shot per minute) will allow for investigation of multi neutron capture processes in laboratory conditions. Parts of the field for astrophysical nucleosynthesis[84,85], such as the investigation of the r-process important for the heavy element synthesis in solar or explosive scenarios become accessible. Especially, the investigation of multi/single neutron capture rates on heavy radioactive or unstable elements is a current challenge and not experimentally accessible in conventionally nuclear physics facilities.

One of the upcoming facilities using radioactive heavy ions is the *Facility for Antiproton and Ion Research* (FAIR) in Darmstadt, Germany[86,87]. The two most limitations for the feasibility of such nuclear astrophysical studies are: 1. the separation of the heavy ion facilities from reactor or accelerator-based neutron sources, and 2. too low neutron fluxes reached by conventional neutron sources. A suitable way that allows such important r-process studies is the combination of laser-driven ultra-high flux neutron sources with future heavy ion facilities like FAIR. Our prospective insights on neutron generation in relativistic laser interactions with long-scale NCD plasmas show a suitable and realistic way to reach conditions for nuclear astrophysical laboratory science using existing high-energy sub-PW laser systems as well as kJ PW-class laser facilities.

## Methods

**Laser and target system.** The experiment was performed at the Petawatt High-Energy Laser for heavy Ion eXperiments (PHELIX) at GSI in Darmstadt, Germany. The Nd:Glas laser amplifier system delivered a s-polarized beam with a central wavelength of 1053 nm and a highest main pulse to nanosecond pre-pulse contrast of more than $10^{11}$. The pulse length was $(0.75 \pm 0.25)$ ps. The laser pulse was focused by an off-axis copper parabolic mirror on the target under $\approx 7°$ to target normal. To conduct a moderate relativistic intense laser pulse of $(1–2.5) \times 10^{19}$ W/cm$^2$ a $(90 \pm 10)$ J pulse was focused by a focal length of 150 cm on the target, where the FWHM of the elliptical focal spot size was $(12 \pm 2)$ μm $\times$ $(18 \pm 2)$ μm. 22% of the laser pulse energy is contained in the FWHM of the focal spot which results in an energy of $(20 \pm 2)$ J on the target. The high intensity laser pulse of $10^{21}$ W/cm$^2$ was produced via a 180 J pulse focused by a parabolic mirror with a focal length of 40 cm. The FWHM of the elliptical focal size was $(2.7 \pm 0.2)$ μm $\times$ $(3.2 \pm 0.2)$ μm, where the laser pulse energy within the FWHM was $(38 \pm 2)$ J. The long focal length was used in laser shots on aerogel-foam target systems and the short focal length was used in shots on foil targets. To realize a NCD plasma a well-defined long pre-pulse with 1.5 ns pulse duration irradiated the foam target system by a delay of 2–3 ns before the main pulse, where the intensity of the pre-pulse was about $5 \times 10^{13}$ W/cm$^2$ and the pulse energy was 1–3 J[55]. For the generation of a sub-mm NCD plasma a plastic aerogel target made from triacetate cellulose, $C_{12}H_{16}O_8$, with a volume density of 2 mg/cm$^3$ and a thickness between 300 μm and 400 μm was used[65]. A fully ionized plasma corresponds to $0.64 \times 10^{21}$ cm$^{-3}$ electron density or 0.64 $n_{\mathrm{cr}}$ ($n_{\mathrm{cr}} = 10^{21}$ cm$^{-3}$). As foil targets, 10 μm thick pure metallic foils (Au, Ti) and 1 mm thick pure Au targets were used.

**Electron diagnostics.** The electron spectrometers were realized using a homogeneous magnetic field of 0.99 T created by two parallel positioned flat neodymium permanent magnets. Calibrated imaging plates (IP) from BASF[88,89] were used as detectors. The spectral resolution was optimized by numerical simulations that consider the measured two-dimensional magnetic field distribution and an entrance slit with a size of 300 μm (width) $\times$ 1 mm (height). The experimental error in the detection of the electrons was not higher than 2%, which allows for a precise spectral measurement in the electron energy range between 1.75 MeV to 100 MeV. To increase the signal-to-noise ratio and to reduce the MeV-bremsstrahlung background, each spectrometer was shielded by a CuW collimator placed in front of the entrance hole. The spectrometers were positioned in laser forward direction at a distance of 405 mm at 0°, 15° and 45° in the horizontal plane perpendicular to the laser polarization direction. Furthermore, a stack of three steel cylinders with a thickness each of 3 mm and a radius of 200 mm were placed in a distance of 230 mm from the laser interaction target. Between the steel cylinder, IPs were fixed to measure the angular distribution of the electron beam in horizontal as well as vertical directions for two different energy ranges above 3 MeV and above 7.5 MeV.

**Nuclear diagnostics for MeV bremsstrahlung and neutrons.** After each laser shot, the nuclear-activated samples were analyzed by a low back ground $\gamma$-spectroscopy using 60% high purity germanium (HPGe) detector systems to identify the reaction channels of the isotopes of interest (Fig. 7). A stack of pure (more than 99%) elemental Au, Ta, Cr, and In plates was used as an activation detector system, where their natural isotope abundance provide different nuclear reaction thresholds (see Fig. 8a). The reaction yield of the activated isotopes in the sample is shown in Fig. 8c, d. In the activation samples, multi-neutron disintegration reactions were observed, triggered by MeV-photons and particles (Fig. 8a, b). The size of each plate was 15 mm $\times$ 15 mm $\times$ 1 mm, while the thickness of the indium foil was of 0.25 mm. The activation detector stacks were placed in a distance of 230 mm at two different angular positions with respect to the laser axis (5° and 15°). The main uncertainty in determination of the reaction yield is caused by the geometrical detector efficiency error, where the geometry and self absorption processes of $\gamma$ decay lines within the activation samples are considered by the error of the counting statistics of the HPGe detector. The nuclear reaction yield of each $i$-th reaction $Y_i$ is the convolution of the initial photon or particle spectrum $N_{\gamma,\mathrm{p}}(E')$ with the energy dependent reaction cross section $\sigma_i(E')$ (see Fig. 8a, b):
$Y_i = N_T \int_{E_{\mathrm{th}}}^{\infty} N_{\gamma,\mathrm{p}}(E')\sigma_i(E')dE'$, where $N_T$ is the surface atomic number density of each sample. The photon or particle spectrum was deconvolved by weighting the reaction yield with the reaction cross section within each energy bin $dE'$ of the reaction $i$ with the reaction threshold $E_{\mathrm{th}}$. The error in the determination of the initial photon or particle spectrum is the combination of the error of the reaction yield and the width of the corresponding nuclear resonance of the nuclear reaction (Fig. 8a, b). All uncertainties are shown as the error bars of the experimental results presented in the corresponding diagrams in this work. The nuclear reaction cross

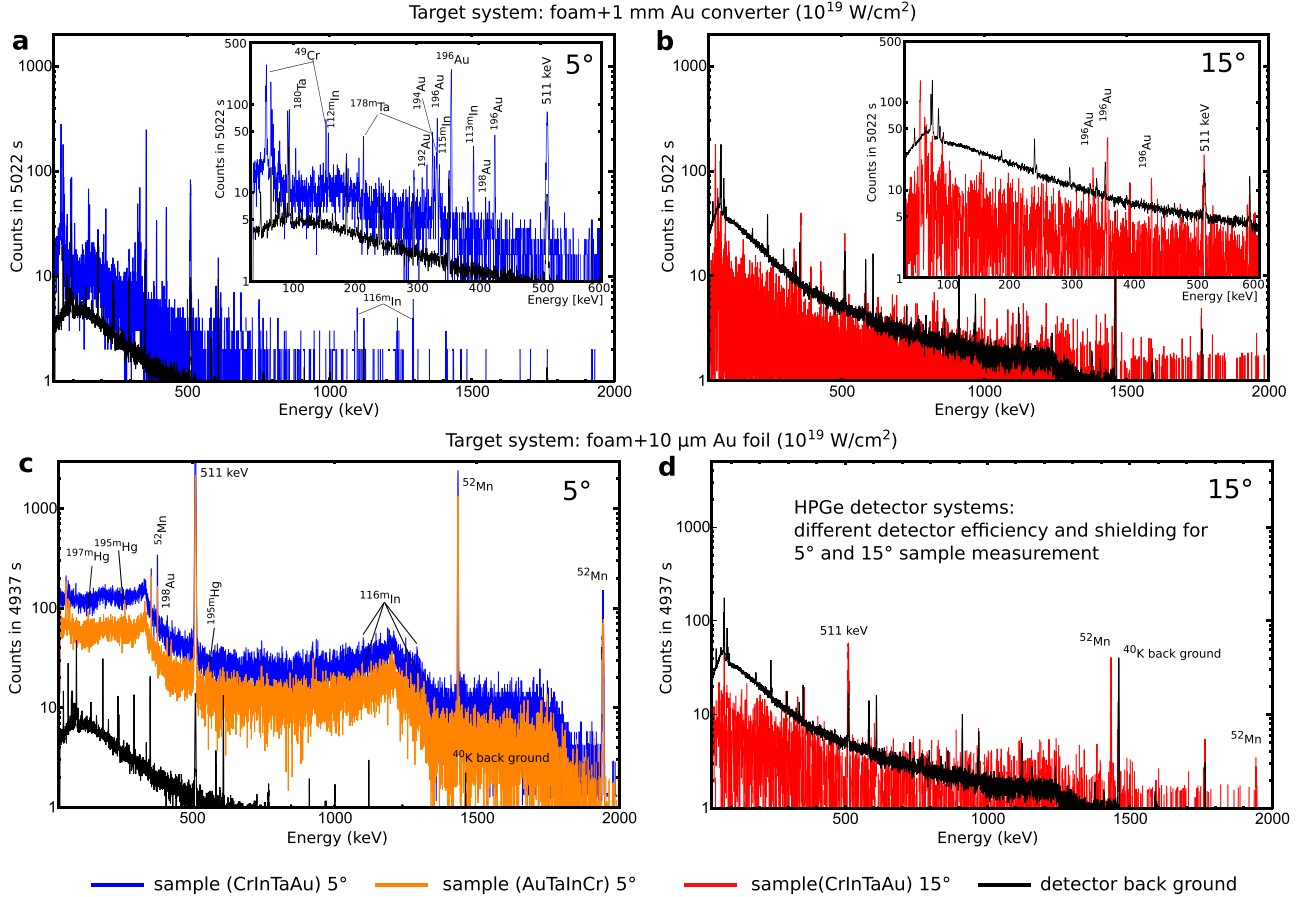

**Fig. 7 Gamma spectrum.** Results of the gamma spectroscopy of the activated samples: **a** and **c** at 5° (blue and orange spectra), **b** and **d** 15° (red spectra) for the laser-target setup scenarios. The black spectrum in each panel represents the detector back ground spectrum. Blue and orange colors are used for different sequences of activation plates in the stack. The unequal count rate of the back ground between the panels **(a)**, **(c)** and **(b)**, **(d)** is caused by the differently HPGe detector stations with its own back ground shielding. Results are shown after subtraction of the detector background. An important indicator for signal differences between 5° and 15° is the pulse height of the annihilation line at 511 keV, because the most activated isotopes are $\beta^+$-emitter.

sections were taken from the EXFOR data base[77] and evaluated by the Talys nuclear reaction code[90].

In the following, the indium activation method is described for the neutron fluence measurement. Reaction channels induced by MeV-photons leading to the same In isotopes with respect to the neutron-induced reactions were also considered. The contribution of these reactions is determined by weighting the experimental MeV-bremsstrahlung spectrum (see Fig. 3) with the cross sections (Fig. 8a, b) of the corresponding reaction channels: $^{113}$In$(\gamma,\gamma')^{113m}$In, $^{113}$In$(\gamma,n)$ $^{112m}$In, $^{115}$In$(\gamma,\gamma')^{115m}$In, $^{115}$In$(\gamma,n)^{114m}$In, $^{115}$In$(\gamma,2n)^{113m}$In, $^{115}$In$(\gamma,3n)^{112m}$In. The total $\gamma$ induced reaction yields of the activated indium isotopes are shown in the right panel of Fig. 8d. For the setups with foam target systems (1a, 1b) the contribution of the $\gamma$ induced indium isotopes $^{113m}$In, $^{112m}$In and $^{114m}$In are compared to the same neutron-induced isotopes. However, the $\gamma$ induced contribution into production of $^{115m}$In is more than one order of magnitude lower compared to the neutron-induced contribution. The neutron-induced reaction yields shown in the left panel of Fig. 8d are determined by subtraction of the corresponding yields of $\gamma$ induced In isotopes from the measured total reaction yields. It has to be noted that the high value of the $^{114m}$In yield is reached via the epithermal neutron capture reactions and the fast neutron scattering processes.

To evaluate a neutron energy distribution, following neutron energy ranges are considered: 1 eV, 2.5 MeV and 14 MeV. This corresponds to three key reaction channels taken into account for the reconstruction of the neutron number: $^{115}$In$(n,\gamma)^{116m}$In, $^{115}$In$(n,n')^{115m}$In and $^{113}$In$(n,2n)^{112m}$In. The indium isomers $^{113m}$In from reactions $^{113}$In$(n,n')^{113m}$In, $^{115}$In$(\gamma,2n)^{113m}$In and $^{113}$In$(\gamma,\gamma')^{113m}$In, are not used: first, because of the high natural abundance of the stable mother isotope $^{115}$In and second, because of the high fluence of $\gamma$'s in the corresponding energy range, where the $^{113m}$In is produced via $\gamma$ induced reactions. Also not used for the reconstruction of neutron numbers is the $^{114m}$In isomer that is produced

via the reaction $^{115}$In$(n,2n)^{114m}$In, because of the activation of $^{114m}$In is: first, dominated by $\gamma$ induced reactions $^{115}$In$(\gamma,n)^{114m}$In (high $\gamma$ fluence at 14 MeV, high abundance of $^{115}$In) and second, by the neutron capture reaction $^{113}$In$(n,\gamma)^{114m}$In with a high reaction cross section of $10^4$ barn in the epithermal neutrons.

For setup 1a, the simulated reaction yield is exemplary shown in Fig. 8d (black stars). The GEANT4 simulation is in a well agreement with the experimental yields of indium isomers (yellow dots in Fig. 8d). The simulation considers reactions generated within the sample stack placed at 5° and 23 cm distance from laser interaction point. The $^{116m}$In isomers can be also generated by neutron scattering processes on environmental materials. Additional GEANT4 simulations of the background neutron number caused by the target chamber environment resulted into less than 30% contribution to the measured indium isomer yields, what is inside the experimental accuracy.

In the present work, the mean neutron kinetic energy is determined by an advanced isotope relation equation, where the basic idea is taken from[75]. Starting from the $^{115}$In mother isotope, the daughter isotopes $^{116m}$In activated by thermal/ epithermal neutrons and the $^{115m}$In activated by fast neutrons are generated. The resonance, excited in the reaction channel $^{115}$In$(n,n')^{115m}$In is centered in the fast neutron region and overlaps the cross section region of the reaction channel $^{115}$In$(n,\gamma)^{116m}$In (Fig. 2 in[75]). Therefore, the cross section ratio of the $^{116m}$In production to the cross section range of $^{115m}$In production is related to the isotope yield ratio of $^{116m}$In to $^{115m}$In: $\frac{\sigma 116_m\text{In}}{\sigma 115_m\text{In}} = \frac{Y 116_m\text{In}}{Y 115_m\text{In}}$. The reaction yield of $^{116m}$In and $^{115m}$In are known from the experiment. Then, the relation equation allows to determine the cross section ratio and therefore the neutron energy where the $^{116m}$In isotopes are produced covered by the resonance range for $^{115m}$In production at $\sigma 115_m\text{In}$ =0.35 barn. This determined cross section range corresponds to the mean kinetic energy $E_{\text{mean}}$ of the initial neutrons.

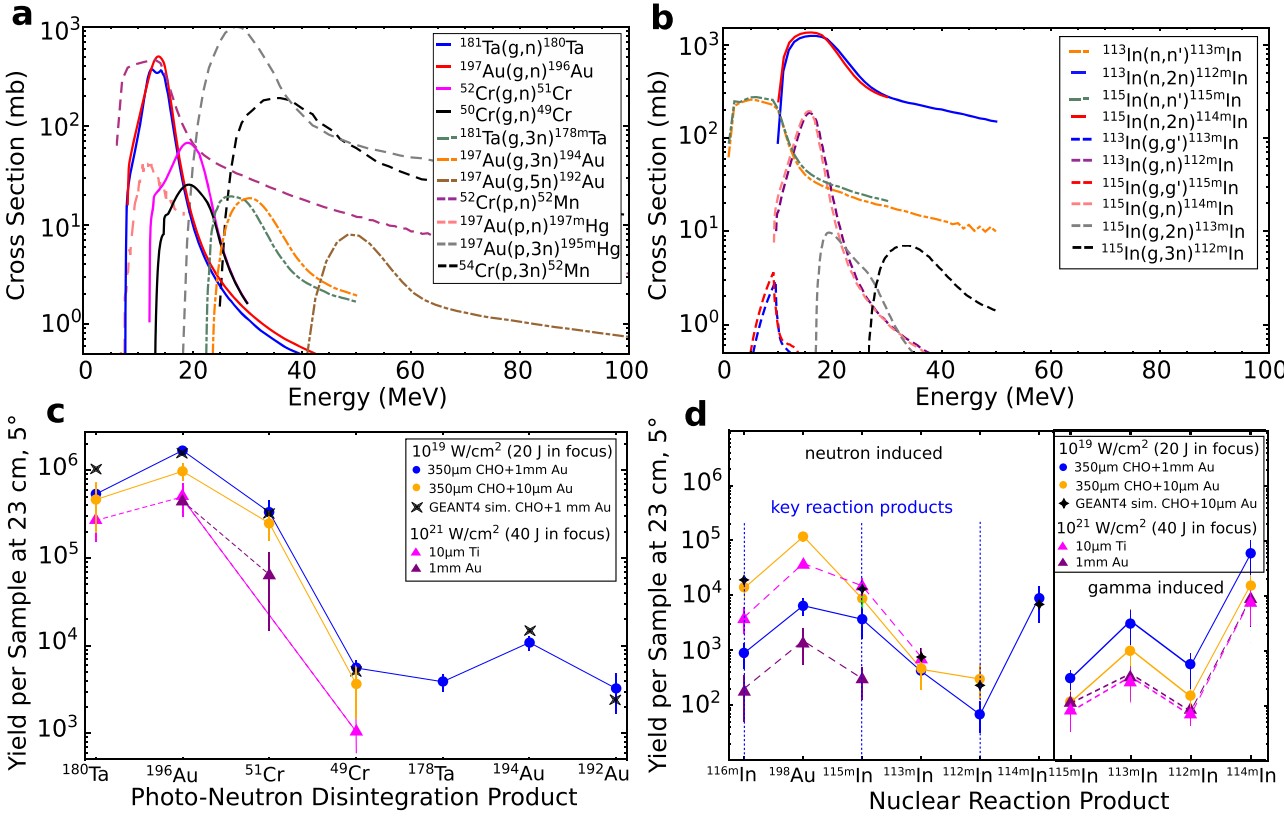

**Fig. 8 Cross section. a** and **b** present the nuclear reaction cross sections for the observed gamma, proton and neutron-induced reactions. **c** show the measured reaction yields together with simulated by means of GEANT4. The error bars represent the total error of the reaction yields, where the main uncertainty is caused by the geometrical detector efficiency error and the error of the counting statistics. In panel **d** on the left, the pure neutron-induced reaction yields are shown for the laser-target setups (these numbers are obtained from the total measured yields after subtraction of yield of the corresponding gamma induced reactions leading to the same isotope). The key reaction products are used for neutron number reconstruction.

## Data availability

The authors declare that the data supporting the findings of this study are available within the paper.

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

## Acknowledgements

The presented results were obtained in the experiment P176 performed in the framework of FAIR Phase-0 at the PHELIX laser facility at the GSI Helmholtzzentrum für Schwerionenforschung GmbH in Darmstadt, Germany. We thank the PHELIX team for the support. Also we thank Mr. Xiaofei Shen for theoretical support. The research of N.E.A. was supported by The Ministry of Science and Higher Education of the Russian Federation (Agreement with Joint Institute for High Temperatures RAS No 075-15-2020-785 dated September 23, 2020).

## Author contributions

M.M.G. was writing the manuscript and conceived the scheme. M.M.G. and P.T. evaluated the results from the nuclear diagnostics. M.M.G. and M.G. performed the illustration of the results. A.S. and A.K. were performing GEANT4 simulations. A.P. and N.E.A. provided theoretical support. N.G.B. fabricated the aerogel-foam targets and provided related target information. O.N.R., S.Z. and M.G. evaluated the results from the electron diagnostics and performed illustrations. The experimental studies were performed by M.M.G., O.N.R., P.T., M.G., and S.Z. All authors discussed the results, commented on the manuscript, and agreed on the contents.

## Funding

## Competing interests

The authors declare no competing interests.
