## [Peer Review File · Nature Communications]

Forward-looking insights in laser-generated ultra-intense γ -ray and neutron sources for nuclear application and scienceREVIEWER COMMENTS

Reviewer #1 (Remarks to the Author):

The manuscript NCOMMS-20-40600 shows a rapid advances in efficient generation of ultra-intense gamma ray and neutron source driven by relativistic laser interactions at intensity of 10^{19} W/cm², which is readily attainable at laser-plasma laboratory. It is a solid extension to the previous studies (see Refs. [53, 54]). However, several major concerns should be addressed before the referee can make a recommendation.

1. INTRODUCTION Section

1) The author(s) summarized shortly the potential of high-brilliant gamma beams to the investigation of photodisintegration reactions in nuclear astrophysics as well as in production of medically interesting radioisotopes. We suggest the author(s) to include the following recent studies, which address exactly the above-mentioned topics: Matter and Radiation at Extremes 4 (2019) 064401; Applied Physics B 122 (2016) 8; Nuclear Science and Techniques 27 (2016) 113 (in production of medical radioisotopes); Physical Review C 98 (2018) 054601 (for investigation of photo-disintegration reactions in nuclear astrophysics).

2) Lines 99-102, the presentation is wrong. The ELI-NP will produces tunable and narrow gamma-ray beams via Compton scattering of (merely) Joule-level, picosecond laser pulse on ~ 700 MeV electron beams. The citation to a more recent publication is suggested.

3) Lines 133-140, a litter more extension should be given here. Besides the single PW laser scheme, the geometry of counter-propagating PW laser pulses are still promising to investigation of nonlinear QED effects and then production of overdense QED plasmas [see Physics of Plasmas 22 (2015) 063112, Scientific Reports 8 (2018) 8400, and reference therein].

2. EXPERIMENT Section

4) In the caption of Fig. 3, the experimental condition for such observation should be specified.

5) Line 355, the value of 1.4% should be checked since the referee calculated the value to be $4.3 \text{ msr}/0.27 \text{ sr} = 1.6\%$.

6) Lines 363, the statement of “a 10 times higher reaction yield”. However, it is shown visibly in Fig. 5(a).

7) Line 390, how did the author(s) obtain the value of photon number, 1.7×10^{11} and the value of energy conversion efficiency, 1.1 – 1.7%? It is obtained by integrating the curve of Fig. 5(c) over the gamma-ray energy higher than 10 MeV?

8) Line 430, there is a typo for $^{113}\text{In}(\gamma, n)^{115\text{m}}\text{In}$. Did the author(s) mean $^{113}\text{In}(\gamma, \gamma')^{113\text{m}}\text{In}$ or $^{113}\text{In}(\gamma, n)^{112\text{m}}\text{In}$?

9) Lines 430-432, the author(s) should present a more detailed analysis to conclude the contribution of the gamma induced reaction yields. In fact, the gamma induced reaction yields would not too small (but as the author(s) stated, it is two orders of magnitude lower), compared to the secondary neutron induced reaction yields. According to a very preliminary estimation, some of them would have comparable or even higher values. The following reaction channels should be considered carefully: $^{113, 115}\text{In}(\gamma, \gamma')^{113\text{m}, 115\text{m}}\text{In}$, $^{115}\text{In}(\gamma, n)^{114\text{m}}\text{In}$, $^{115}\text{In}(\gamma, 2n)^{113\text{m}}\text{In}$, due to both the high-flux of primary gamma rays (of the order of 10^9 at 10 MeV), whereas these reactions have smaller reaction cross sections compared to fast neutron scattering cross sections.

10) Lines 447-448, the statements (and the following ones) should be revised since the indium isotopes are produced not only by neutron-induced reactions, but also by energetic gamma-induced reactions. It should be noted that in photo-neutron reactions, the secondary neutrons are mainly fast neutrons and the epithermal neutrons produced are scarce. As a result, the neutron energy spectrum should be employed as REAL as possible, to estimate further the neutron-induced reactions on indium isotopes.

11) According to the experimental setup and conditions, the $^{113}\text{In}(\gamma, n)^{112\text{m}}\text{In}$, $^{115}\text{In}(\gamma, 3n)^{112\text{m}}\text{In}$, and $^{113}\text{In}(n, 2n)^{112\text{m}}\text{In}$ are possibly main reaction channels to produce the $^{112\text{m}}\text{In}$ isomers. The $^{114\text{m}}\text{In}$ isomers can be resulted from the following reaction channels, $^{115}\text{In}(\gamma, n)^{114\text{m}}\text{In}$, $^{113}\text{In}(n, \gamma)^{114\text{m}}\text{In}$, and $^{115}\text{In}(n, 2n)^{114\text{m}}\text{In}$. The author(s) should identify which reaction channel(s) play a key role in the production of indium isomers. For production of $^{113\text{m}}\text{In}$ isomer, the contribution of reaction $^{115}\text{In}(\gamma, 2n)^{113\text{m}}\text{In}$ should be considered due to the high abundance of ^{115}In and high-flux gamma rays at ~20 MeV.

12) The referee agreed to that the $^{116\text{m}}\text{In}$, ^{198}Au , and $^{115\text{m}}\text{In}$ were produced mainly by (n, gamma) reactions or (n, n) reaction, in which the neutron generation may be dominated by proton driven nuclear reactions. However, the other indium isomers shown in Fig. 5(b) can be produced not only by neutron-induced reactions but also gamma-induced reactions. Fig. 5(a) only shows the reaction products induced by neutrons, but neglects those induced by energetic gamma rays. How did the author(s) substrate the contribution of gamma-induced reaction yields? In addition, it is waiting for more explanation on the data variation shown in Fig. 5. For example, in the scenario of aerogel-foal +

10 μ m Au target irradiated by 10^{19} W/cm² laser pulse, the author(s) did not show the yield for ^{113m}In . In the scenario of aerogel-foal + 1mm Au target irradiated by 10^{19} W/cm² laser pulse, why is the yield of ^{116m}In comparable to the yield of ^{114m}In and why is the yield of ^{112m}In 20-30 times lower than the yield of ^{114m}In ?

13) A detailed expression to describe the curves shown in Fig. 5(d) should be provided.

14) In laser interactions with aerogel-foal target plus 1 mm thick high-Z radiators, a large number of energetic gamma rays are produced (see Fig. 4), which can excite sufficiently photon-induced nuclear reactions, producing these indium isomers. It is hard to justify the validation of the contexts labeled in Lines 460-464.

15) In Fig. 3(c), it lacks of showing the possible reaction cross sections for indium isomer production with energetic gamma rays.

16) Line 546, a typo 'variing'

17) Similar to Fig. 7 showing the comparison of experimental photo-neutron disintegration reactions yields (see Fig. 5(a)) with simulated ones, it is necessary to reproduce with GEANT 4 toolkit the reaction product distributions shown in Fig. 5(b).

18) What is the condition for the optimized case shown in Fig. 7(c)?

19) Line 673, it is confusing to obtain the neutron fluence of > 2 times 10^{11} n/cm². I guess this is not the experimental result but an estimated one with optimized condition.

20) It is better if the author(s) could provide one or two figures to show the difference of spectral and spatial patterns between photon and proton induced neutrons.

Reviewer #2 (Remarks to the Author):

The manuscript “New Insight in laser-generated ultra-intense gamma and neutron sources for nuclear applications and science” by Gunther et al. presents the experimental results using NCD targets to enhance electron acceleration, which leads to higher (an order of magnitude) gamma and neutron generation compared to flat-foil targets. The authors estimate the gamma and neutron yield using activation measurements, with some relevant fittings. The paper is written adequately well and quite detailed in terms of the technical aspects.

My main concern is that the paper does not provide the sort of fundamental or general advance in understanding in physics over previously published work in this field that would excite the interest of a wide, non-specialist audience of physicists. Furthermore, the neutron and gamma yields reported in the paper does not really show fundamentally record-high figures, that has not been demonstrated with other possible mechanisms.

As mentioned above, the paper is quite strong from a technical point of view; however, it lacks from the physics side. The enhancement of gamma flux is presumed by the direct laser acceleration of the electrons in a near-critical density plasma, a topic which has been studied by many groups over the years with many interesting results, not very unlike the results presented here. It is possible that the interaction condition may have been somehow better optimised, or there is a new mechanism at play (for instance, possible enhancement of laser intensity in the NCD plasma due to self-focussing, which has been suggested by other groups studying similar interaction conditions), which has led to more effective electron acceleration. However, there have been no in-depth discussions or simulations looking at the core mechanism, which could have been more meaningful for the scientific community, rather than the fine details about the diagnostics or cross-sections presented in the manuscript.

Regarding the experiment, I am also a bit surprised that there was no secondary diagnostic to reaffirm or support the claims made. The activation measurements, by principle, suffers from large uncertainties unless there are good statistics. Furthermore, exposing simultaneously multiple broadband sources (gamma, electron and neutron) to a stack of different material is fundamentally too convoluted to provide a reliable estimation about the different sources. For instance, the neutron activation cross-section varies by several orders of magnitude going from MeV to eV. The activation foils, being a time-integrated detector, register neutrons multiple times, depending on the surrounding and shielding, while the neutrons are scattered and moderated by the chamber walls, nearby objects etc. Without any complementary measurements, I would have reservation believing the estimations any better than an order of magnitude.

I am also unsure how valid are authors’ claims regarding the neutron and gamma fluxes compared to the state-of-art. For instance, the neutron flux is compared with only a few selected papers and not in terms of flux per steradian. There are recent papers showing neutron fluxes close to 10^{11} n/sr, one of which is from the same laser facility. The similar shortcoming I found for the reported gamma fluxes, where the claims are not sufficiently founded.

Finally, I have some concerns about the author's projection for nucleosynthesis application, which they have focussed mainly in this paper as a near-term possibility. There has been a significant mix-up between the neutron flux, average neutron flux, whether fast or thermal flux... while comparing with the conventional sources. Furthermore, the r- and s- process requires keV neutrons, where the (p,n) and (gamma,n) sources are of MeV energies. Hence, we will lose 3-4 orders in neutron flux while moderating the fast neutrons. The fast neutrons are indeed of sub-ns or ps duration. However, the moderation will make the neutron bursts of microsecond duration, i.e. we again lose three orders in brightness. Hence I would be cautious making strong claims of meeting the required flux based on a crude scaling/estimation.

Certainly, the data presented in this paper show high promise, and I congratulate the authors for their fine work. Although I don't believe the paper meets the requirement to be published in 'Nature communication', I would strongly recommend this paper for a more specialised journal, such as Applied physics letters.

Reviewer #3 (Remarks to the Author):

Please find my comments in the attached PDF.

In general I think this study has merit and is well suited to Nature Comms, however requires additional polish prior to publication. I have a listed some comments as they appear throughout the text and a few general comments that should be addressed before recommending for publication in Nature Comms. The line numbers are given as they appeared on the proof.

[Reviewer #3 attached PDF available below.]

The authors present a series of detailed measurements on the gamma, and neutron production from near-critical density (NCD) foams and converter targets, compared to standard foil targets. The authors compare intensities of 10^{19} W/cm² on the NCD foams and 10^{21} W/cm² on the solid foils. The NCD foams drive a DLA process to accelerate a high charge high energy electron beam that is then measured by a variety of processes, including spectro-spatial measurements of the emitted electrons, measurement of reemitted activation lines in materials, and then comparisons with monte-carlo simulations. The NCD plasma is seeded by a nanosecond laser pulse, followed by the main driving pulse, the foams are used provide a long path length with a semi-homogenous plasma density. The key findings here are the both the level of detail in the measurements and the high conversion efficiency between both laser-x-ray and laser-neutron sources in comparison to other measurements in the field. This makes it of clear interest to both others in the laser-plasma field and the wider secondary source community. The work is a good demonstration of DLA processes in experimental conditions, the utilisation of many activation lines measured in turn by a high purity germanium detector provide an un-ambiguous measurement of the radiation the authors describe.

Generally, I think the paper needs polish rather than substantial changes to aid the readers and to highlight the results within. There should, also, be significantly more discussion on the prior work done with NCD foam targets as there is currently more discussion on how other techniques of generating super-ponderomotive electrons work but very little discussion on prior work with foams. Particular mention to Willingale *et al* [1] who presents a similar discussion into NCD foams lending themselves to DLA processes within the plasma channel and Nakamura *et al* [2] who are using thinner foams but demonstrate enhancement without pre-igniting the under-dense channel.

I have a listed some comments below as they appear throughout the text and a few general comments for the authors, I feel that these should be addressed before recommending for publication in Nature Comms. The line numbers are given as they appeared on the proof.

[1] <https://iopscience.iop.org/article/10.1088/1367-2630/aae034/pdf>

[2] <https://aip.scitation.org/doi/pdf/10.1063/1.3507294>

General:

- Discussion into the prior NCD foam work already studied in the field, while this paper adds significant detail and discussion on the emission from these targets it is worth acknowledging the prior work in the field, and discussing their relevance herein.
- While important to highlight the activation lines and actual measurement of the germanium signal the data is currently presented unclearly. For example Figure 5 shows first the yield from the activation measurements and then separates the fluence as a function of energy, what matters from a scientific perspective is the second half of the graph. My suggestion is to simplify these figures to only show the scientific aspect rather than “showing the working”, else the reader needs to first look at Fig 5a) then compare to Fig 3c) to understand what point is trying to be made. The same is true in Fig 7.

Abstract:

- I think a key result is the conversion efficiency for laser-neutrons this should be included in the findings in the abstract alongside the absolute numbers

Introduction:

- Conversion to N/cm² s requires a further discussion on the temporal profile of the neutrons in this process, you mention how for the S and R process it must be less than a second and while the neutron pulse is certainly shorter than this when normalising to N/ cm² s it matters as to what pulse length you are using.
- L102, “planned” is misspelled
- L124, I think counter rather than “contra”
- L138, remove very. I dislike the emotive language and think it would be better to state that these conditions enable QED studies.
- L159, what is multi 10th?

Experiment:

- L205, I think you should state that the different intensity regimes are achieved by different focussing conditions. It's stated in the methods but it is important here for context.
- L210, was the plasma density measured at all?
- Fig1, sentence structure in the caption needs work.

Results and Discussion:

- L246, throughout I think target is clearer than radiator for the converter foil.
- L253-256, This description is verbose, in line with my general comments above, I think some of the findings in the paper would be better presented as figures with clear results. For instance for this passage a plot of angle versus the two measured components in both target cases would show the variation much better than the written description.
- L262-264, this process is stochastic by nature and one would expect greater variation from this. Adding a caveat to the effect of “the mean free path is ..” would help here.
- Fig 2, why is the thin target for 10²¹ Ti rather than Au, it is hard to do the direct comparison you are after with such a difference in Z and Density.
- Fig 2, the caption is ordered wrong with (b) last..

- Fig 2, in general I think it wise to smooth the raw electron spectra plots to avoid the very busy structure we see here. The core information you wish to present here is the increase between the conditions this can be better expressed by varying the parameter on the bottom (intensity, or angle of measurement) against the temperature or flux.
- L290, How are you defining maximum energy? In both Fig 2c and d there are dots above 40 MeV for each colour,
- Fig 3, while this data is clearly important to how the measurement was done what the reader cares about is the cross-sections in Figure 3c and what the measured signal tells you about the photon/neutron spectra not the raw counts from the germanium. It makes sense to include this as part of the methods description but not the core of the paper.
- L335, given the nature of the paper being a comparison between DLA with foams and the higher intensity solid target interaction you should compare the emission cone between these two cases not the two foam shots. Do you have the emission cone data for the solid target as well?
- Fig 4 (+surrounding discussion), how did you inject the electron beam into the target in GEANT? As this will influence the presented result.
- Fig 5, This should be reduced to just panels C and D, I don't think the upper panels add anything as the reader must jump back to the cross-sectional data to determine what these lines really mean. What is important here is the flux/temperature presented in the second half. It would also be worth including the uncertainty in energy associated with the cross-sections
- L470, "a twice"
- L552-555, it is unclear as to if this is an incredibly directional beam with $\leq 5\text{msr}$ divergence or the peak was measured over that angular range.
- L564-566, a comparison to solid targets here is important.
- L568, "unfortunately" and "scaling"
- L570, can the proton beam divergence be estimated from other work in the field? Again these foams are not unique and have been used to drive proton emission is there a comparison that can be made to aid this discussion?
- L576, emotive language
- Table 1, this is hard to read with the multiple line entries for the row labels, this should be formatted differently to make it simpler. Either stretching it across the whole page or adding in lines.
- Fig 7a, again irrelevant as Fig 7c distils the important information for the reader.
- Fig 7b, I like this schematic, the description needs to be expanded but it could stand with D as its own figure.

Conclusions

- L710, the conversion efficiency I read as the main highlight from Table 1 this should be added into the abstract.

Methods

- L768, what caused the spot to be so elliptical? And is the energy in the wings of the spot detrimental to the plasma channel in anyway?
- L775, don't use about, state the uncertainty.
- L785, was the density measured at all? Is the assumption of fully ionised foam valid for the conditions?
- L789, titanium. Was the thin foil with the foam target also Ti or was it Au.
- L800, "height"

- L806, delete “massive”.
- L835, it is important here (as mentioned with Figure 5) it is important to discuss the associated errors/uncertainty with this method as you are then fitting to it to find a temperature.

Response to reviewer #1

I Introduction

1) The author(s) summarized shortly the potential of high-brilliant gamma beams to the investigation of photodisintegration reactions in nuclear astrophysics as well as in production of medically interesting radioisotopes. We suggest the author(s) to include the following recent studies, which address exactly the above-mentioned topics: Matter and Radiation at Extremes 4 (2019) 064401; Applied Physics B 122 (2016) 8; Nuclear Science and Techniques 27 (2016) 113 (in production of medical radioisotopes); Physical Review C 98 (2018) 054601 (for investigation of photo-disintegration reactions in nuclear astrophysics).

Answer: In L37 of the revision we added: "...photo-disintegration reactions in nuclear astrophysics [Lan et al., Phys. Rev. C 98, 054601 (2018); instead of Matei et al.]" "...efficient production of medical radioisotopes...[Habs et al., Köster et al., and additional Luo et al., Appl. Phys B 122 (2016) 8, MRE 4 (2019) 064401; Loveless et al. (2019)]."

2) Lines 99-102, the presentation is wrong. The ELI-NP will produce tunable and narrow gamma-ray beams via Compton scattering of (merely) Joule-level, picosecond laser pulse on ~700 MeV electron beams. The citation to a more recent publication is suggested.

Answer: From L101 of the revision, we corrected as follows: "Upcoming high-brilliance and high-intensity gamma beam facilities like the Variable-Energy Gamma Ray system (VEGA) at ELI-NP will produce tunable and narrow gamma beams via Compton backscattering of 100 TW laser pulses on relativistic electrons with energies above 700 MeV [Tanaka et al., MRE 2020]."

3) Lines 133-140, a litter more extension should be given here. Besides the single PW laser scheme, the geometry of counterpropagating PW laser pulses is still promising to investigation of nonlinear QED effects and then production of overdense QED plasmas [see Physics of Plasmas 22 (2015) 063112, Scientific Reports 8 (2018) 8400, and reference therein].

Answer: From L140, we added by the reviewers suggested reference. "...Such new single laser schemes..." "...Also, simulation studies considering a counter propagating multi-PW laser scheme for two-side irradiation have shown promising results in the production of enhanced electron-positron plasma density for strong-field QED investigations [Luo et al., PoP 22, 063112 (2015)]."

II Experiment

4) In the caption of Fig. 3, the experimental condition for such observation should be specified.

Answer: We moved data presented in Fig.3 (gamma spectroscopy), Sec. III (Results and Discussion) to Section V Methods C (Fig.7 revised version). The experimental conditions are

now specified in more details. In the new Fig.8 a, b, c, d, the reaction cross sections and measured and simulated reaction yields are shown.

We hope it will lead to better understanding of the used diagnostics as well as the results shown in the new fig. 3, Sec. III (bremsstrahlung spectra) and fig. 6, Sec. III (neutron spectra) for different laser-target setups.

5) Line 355, the value of 1.4% should be checked since the referee calculated the value to be $4.3 \text{ msr}/0.27 \text{ sr} = 1.6\%$.

Answer: We corrected the typo. The right value is 1.6%.

6) Lines 363, the statement of “a 10 times higher reaction yield”. However, it is shown visibly in Fig. 5(a).

Answer: We major revised the section MeV gamma beam and show the reaction yields in Fig.8, Methods C. Here, one can see, that in the case of setup 1b, the reaction yields are 4 to 10 times higher than in 2b.

7) Line 390, how did the author(s) obtain the value of photon number, 1.7×10^{11} and the value of energy conversion efficiency, 1.1 – 1.7%? It is obtained by integrating the curve of Fig. 5(c) over the gamma-ray energy higher than 10 MeV?

Answer: The number of photons per sr was obtained by integrating the curve Fig.3 (new version). The evaluation of the conversion efficiency is given from L398 (discussion of solid angles for electron and gamma beams) and from L431 (discussion of the conversion efficiency).

8) Line 430, there is a typo for $^{113}\text{In}(\gamma, n)^{115\text{m}}\text{In}$. Did the author(s) mean $^{113}\text{In}(\gamma, \gamma)^{113\text{m}}\text{In}$ or $^{113}\text{In}(\gamma, n)^{112\text{m}}\text{In}$?

Answer: Indeed, this is a typo. The authors mean the reaction $^{113}\text{In}(\gamma, n)^{112\text{m}}\text{In}$.

9) Lines 430-432, the author(s) should present a more detailed analysis to conclude the contribution of the gamma induced reaction yields. In fact, the gamma induced reaction yields would not too small (but as the author(s) stated, it is two orders of magnitude lower), compared to the secondary neutron induced reaction yields. According to a very preliminary estimation, some of them would have comparable or even higher values. The following reaction channels should be considered carefully: $^{113, 115}\text{In}(\gamma, \gamma)^{113\text{m}, 115\text{m}}\text{In}$, $^{115}\text{In}(\gamma, n)^{114\text{m}}\text{In}$, $^{115}\text{In}(\gamma, 2n)^{113\text{m}}\text{In}$, due to both the high-flux of primary gamma rays (of the order of 10^9 at 10 MeV), whereas these reactions have smaller reaction cross sections compared to fast neutron scattering cross sections.

Answer: We discuss indium activations by gammas and neutrons in Sec. Methods C after L885. In this section, the new Fig. 8 presents the cross sections of gamma and neutron induced reaction channels together with the reaction yields separated in pure neutron and pure gamma induced reactions (Fig.8d).

10) Lines 447-448, the statements (and the following ones) should be revised since the indium isotopes are produced not only by neutron-induced reactions, but also by energetic gamma-induced reactions. It should be noted that in photo-neutron reactions, the secondary neutrons are mainly fast neutrons and the epithermal neutrons produced are scarce. As a result, the neutron energy spectrum should be employed as REAL as possible, to estimate further the neutron-induced reactions on indium isotopes.

Answer: In the revised version from L537 it is written:

In this procedure, an input of γ -driven nuclear reactions leading to the same indium isotope type was taken into account (see Fig. 8d).

Further from L540:

In the study of reactions triggered by epithermal neutrons, the strong nuclear resonance with a reaction cross section of 10^4 barn at 1 eV for the neutron capture reaction $^{115}\text{In}(n,\gamma)^{116\text{m}}\text{In}$ was considered. Fast neutrons were investigated by the pure neutron activation channel $^{115}\text{In}(n,n')^{115\text{m}}\text{In}$ with 0.343 barn at 2.5 MeV and the activation channels $^{113}\text{In}(n,n')^{113\text{m}}\text{In}$ with 0.2 barn at 2.5 MeV as well as the $^{113}\text{In}(n,2n)^{112\text{m}}\text{In}$ reaction with 1.5 barn at 14 MeV initial neutron energy. The reaction channels $^{115}\text{In}(n,2n)^{114\text{m}}\text{In}$ and $^{113}\text{In}(n,\gamma)^{114\text{m}}\text{In}$ are not useful because the daughter isotope becomes the same for fast and slow neutrons.

For more discussion, see Methods.

11) According to the experimental setup and conditions, the $^{113}\text{In}(\gamma, n)^{112\text{m}}\text{In}$, $^{115}\text{In}(\gamma, 3n)^{112\text{m}}\text{In}$, and $^{113}\text{In}(n, 2n)^{112\text{m}}\text{In}$ are possibly main reaction channels to produce the $^{112\text{m}}\text{In}$ isomers. The $^{114\text{m}}\text{In}$ isomers can be resulted from the following reaction channels, $^{115}\text{In}(\gamma, n)^{114\text{m}}\text{In}$, $^{113}\text{In}(n, \gamma)^{114\text{m}}\text{In}$, and $^{115}\text{In}(n, 2n)^{114\text{m}}\text{In}$. The author(s) should identify which reaction channel(s) play a key role in the production of indium isomers. For production of $^{113\text{m}}\text{In}$ isomer, the contribution of reaction $^{115}\text{In}(\gamma, 2n)^{113\text{m}}\text{In}$ should be considered due to the high abundance of ^{115}In and high-flux gamma rays at ~ 20 MeV.

Answer: As it is written in the response to item 9), we present from L 885 a more detailed discussion and explanation in the section Methods C. Here we discuss the key reaction channels leading to the indium isomers contributed by gammas and neutrons (see Fig.8d) according to the two different laser-target setups.

12) The referee agreed to that the $^{116\text{m}}\text{In}$, ^{198}Au , and $^{115\text{m}}\text{In}$ were produced mainly by (n, γ) reactions or (n, n) reaction, in which the neutron generation may be dominated by proton driven nuclear reactions. However, the other indium isomers shown in Fig. 5(b) can be produced not only by neutron induced reactions but also gamma-induced reactions. Fig. 5(a) only shows the reaction products induced by neutrons, but neglects those induced by energetic gamma rays. How did the author(s) substrate the contribution of gamma-induced reaction yields? In addition, it is waiting for more explanation on the data variation shown in Fig. 5. For example, in the scenario of aerogel-foal + $10\ \mu\text{m}$ Au target irradiated by 10^{19} W/cm² laser pulse, the author(s) did not show the yield for $^{113\text{m}}\text{In}$. In the scenario of aerogel foal + 1mm Au target irradiated by 10^{19} W/cm² laser pulse, why is the yield of $^{116\text{m}}\text{In}$ comparable to the yield of $^{114\text{m}}\text{In}$ and why is the yield of $^{112\text{m}}\text{In}$ 20-30 times lower than the yield of $^{114\text{m}}\text{In}$?

Answer: As described in item 9, we added in the right panel of the new fig. 8d the gamma induced reaction yields of indium isomers including $^{113\text{m}}\text{In}$.

- the neutron induced reaction yields of indium isomers are determined by subtraction of the same indium isomers from gamma induced reactions, obtained by weighting the experimental bremsstrahlung spectrum (Fig.3) with the cross sections.

- As detailed in the section Methods C, the ^{114m}In is produced by gamma induced reactions, fast neutron scattering and neutron capture reactions: 1. the gamma induced reactions are dominant as shown in fig. 8d. Here, the reaction starts from ^{115}In . 2. the fast neutron scattering process has a high cross section for reactions starting from ^{115}In , also. The natural abundance of ^{115}In is 96%. Additionally, there is a high reaction cross section for the capture reactions. These processes and the high natural abundance of ^{115}In leads to a higher yield in ^{114m}In compared to ^{112m}In . ^{112m}In is produced in photon and neutron induced reaction channels starting from the ^{113}In isotope with a much smaller natural abundance than ^{115}In . These leads to the difference in the reaction yield of ^{114m}In .

13) A detailed expression to describe the curves shown in Fig. 5(d) should be provided.

Answer: We added detailed explanations of new Fig.6 in the neutron section from L565 and in the section Methods C from L942.

14) In laser interactions with aerogel-foal target plus 1 mm thick high-Z radiators, a large number of energetic gamma rays are produced (see Fig. 4), which can excite sufficiently photon-induced nuclear reactions, producing these indium isomers. It is hard to justify the validation of the contexts labeled in Lines 460-464.

Answer: As it is already mentioned, the corresponding discussion is presented in section Neutrons from L523 and Methods C of the revised version.

15) In Fig. 3(c), it lacks of showing the possible reaction cross sections for indium isomer production with energetic gamma rays.

Answer: In the revised version, these cross sections are shown in Fig.8b, Methods C.

16) Line 546, a typo 'variing'

Answer: We corrected the typo to "varying".

17) Similar to Fig. 7 showing the comparison of experimental photo-neutron disintegration reactions yields (see Fig. 5(a)) with simulated ones, it is necessary to reproduce with GEANT 4 toolkit the reaction product distributions shown in Fig. 5(b).

Answer: In the revised manuscript, a comparison of experimental yields of gamma and neutron driven nuclear reactions are shown in Fig.8 c, d, Methods from L928.

18) What is the condition for the optimized case shown in Fig. 7(c)?

Answer: Due to a large amount of information presented in our work, we decided to remove this section and to use the results together with more detailed simulation study in the frame work of a separate paper. In the revision, the optimization is shortly discussed from L742:

GEANT4 Monte Carlo simulations were performed to optimize the neutron production for PHELIX parameters (10^{19} W/cm², EFWHM= 20 J, setup 1b). After placing the Au converter direct to the foam rear-side and increasing converter thickness up to 5-7 mm, a record neutron

fluence of $2 \times 10^{11} \text{ cm}^{-2}$ and a flux of $10^{22} \text{ cm}^{-2} \text{ s}^{-1}$ estimated for ~ 20 ps neutron pulse duration was achieved.

19) Line 673, it is confusing to obtain the neutron fluence of $> 2 \times 10^{11} \text{ n/cm}^2$. I guess this is not the experimental result but an estimated one with optimized condition.

Answer: Indeed, this is a result of simulations for the optimized target geometry.

20) It is better if the author(s) could provide one or two figures to show the difference of spectral and spatial patterns between photon and proton induced neutrons.

Answer: The spectral difference between proton and gamma-induced neutrons is discussed in section "Neutrons" from L523 and is shown in the revised version in the new fig.6.

Concerning optimized target geometry, see note in item 18 and 19.

Response to reviewer #2

R2(1): My main concern is that the paper does not provide the sort of fundamental or general advance in understanding in physics over previously published work in this field that would excite the interest of a wide, non-specialist audience of physicists. Furthermore, the neutron and gamma yields reported in the paper does not really show fundamentally record-high figures, that has not been demonstrated with other possible mechanisms.

Answer 1: At this point, authors opinion is different from those of Reviewer.

The gamma yield (1.7×10^{11} ph for $E > 10$ MeV at 20J, 10^{19} W/cm^2) and the gamma conversion efficiency ($> 1.4\%$) are record values for ICF relevant PW laser systems. The obtained laser-to-gamma conversion is at least 5x higher than the conversion efficiency on the ARC-laser at NIF (0.3%) currently up-graded using compound parabolic concentrator [NIF-report: <https://lasers.llnl.gov/news/nif-creates-matter-and-antimatter-from-light>]. Also, comparison of our experimental results with theoretically predicted conversion efficiency of $\leq 1\%$ in the case of a non-linear Compton backscattering scheme driven by a PW short laser pulse at $> 5 \times 10^{21} \text{ W/cm}^2$ in a NCD plasma coupled with a plasma mirror, demonstrates the advantages of our concept.

From L422 of the revised version of the manuscript, it is written:

Based on these results, the estimated total number of photons > 10 MeV reaches 1.7×10^{11} . Taking into account the effective temperature of (12.7 ± 2.1) MeV and laser energy of 20 J, we end up with an ultra-high conversion efficiency of the laser energy into > 10 MeV gammas of $(1.4 \pm 0.3)\%$. Compared to the present work, a new concept of a non-linear Compton backscattering scheme driven by a PW short laser pulse at $> 5 \times 10^{21} \text{ W/cm}^2$ in a NCD plasma coupled with a plasma mirror was recently discussed in [65]. The described simulation studies show a broad-band spectrum that contains $> 10^{11}$ photons with energies above 10 MeV with less than 1% laser-to- energy conversion efficiency.

[65] Huang, T. W. et al. Highly efficient laser-driven Compton gamma-ray source. New Journal of Physics 21, 013008 (2019).

Ultra-high laser-gamma conversion efficiency obtained in our work leads to a record efficiency of the neutron generation in gamma-driven nuclear reactions (see table 1, page 9).

Moreover, the laser-gamma conversion efficiency obtained using sub-mm long pre-ionized foam targets, although it happens firstly via the laser energy transfer to electrons, is higher compared to laser-ion conv. eff. reported [46, 47] and summarized in table 1.

Tremendous achievement of the presented work is that all these record values are obtained at moderate relativistic laser intensities and are applicable for current available kJ-class PW laser systems.

R2 (2): As mentioned above, the paper is quite strong from a technical point of view; however, it lacks from the physics side. The enhancement of gamma flux is presumed by the direct laser acceleration of the electrons in a near-critical density plasma, a topic which has been studied by many groups over the years with many interesting results, not very unlike the results presented here. It is possible that the interaction condition may have been somehow better optimized, or there is a new mechanism at play (for instance, possible enhancement of laser intensity in the NCD plasma due to self-focussing, which has been suggested by other groups studying similar interaction conditions), which has led to more effective electron acceleration. However, there have been no in-depth discussions or simulations looking at the core mechanism, which could have been more meaningful for the scientific community, rather than the fine details about the diagnostics or cross-sections presented in the manuscript.

Answer 2: We thank the Reviewer for this comment and added additional details to the DLA process in the section 2 (Experiment). In this work, we did not present extended theoretical analyses of the DLA-process in sub-mm long NCD plasma, since it was already done for discussed laser-target parameters in currently published paper [54, PPCF 2020, Rosmej et] and other previous theoretical works devoted to experiments at PHELIX [60, 61]. In addition, we discuss the difference between our approach and a multilayer target concept used to enhance the proton production at ultra-relativistic laser intensities.

In revised version, in line 225 it is written:

The effective temperature of the DLA accelerated electrons exceeds more than one order of magnitude the ponderomotive potential and their energies extend up to 100 MeV, already at moderate relativistic laser intensities. [53, 54]. The reason for this behavior is a long acceleration path in a NCD plasma ensured by pre-ionized sub-mm thick foams and a relatively long sub-ps laser pulse duration. This differs from the case of the so-called multilayer targets where a low-density layer of a few-micron thickness is added on the illuminated side of a thin, high-density layer. Such kind of a target was used in the last decade for proton acceleration at ultra-relativistic intensities and sub 100-fs pulses [69-71]. In [70], the generation of hot electrons from a double layer target irradiated by a laser pulse with an intensity of 2×10^{20} W/cm² was reported. The target consisted of a several micrometers thin NCD plasma slab ($0.5 n_{cr}$) an over-dense foil. The measured effective electron temperature was of 8 MeV, what is only twice higher than the ponderomotive potential.

R2 (3): Regarding the experiment, I am also a bit surprised that there was no secondary diagnostic to reaffirm or support the claims made. The activation measurements, by principle, suffers from large uncertainties unless there are good statistics. Furthermore, exposing simultaneously multiple broadband sources (gamma, electron and neutron) to a stack of different material is fundamentally too convoluted to provide a reliable estimation about the different sources. For instance, the neutron activation cross-section varies by several orders of magnitude going from MeV to eV. The activation foils, being a time-integrated detector, register neutrons multiple times, depending on the surrounding and shielding, while the neutrons are scattered and moderated by the chamber walls, nearby objects etc. Without any complementary measurements, I would have reservation believing the estimations any better than an order of magnitude.

Answer 3: We agree with the concern of the Reviewer that the application of the secondary diagnostics would increase the credibility of our results. In the upcoming beam-time, we plan to use additional diagnostics such as bubble detector systems besides others.

Nevertheless, we are confident about the reliability of our measurements supported by MC simulations, which demonstrate a good agreement between experimental and simulated reaction yields (see fig. 8c,d, methods). The GEANT4 simulations that considered the geometry of the experimental set-up and the measured electron energy and angular distributions presented in Fig.2, showed very low effect of the surrounding.

This is pointed out in line 924, methods:

Additional GEANT4 simulations of the background neutron number resulted into less than 30% contribution to the measured indium isomer yields, what is inside the experimental accuracy.

R2 (4): I am also unsure how valid are authors' claims regarding the neutron and gamma fluxes compared to the state-of-art. For instance, the neutron flux is compared with only a few selected papers and not in terms of flux per steradian. There are recent papers showing neutron fluxes close to 10^{11} n/sr, one of which is from the same laser facility. The similar shortcoming I found for the reported gamma fluxes, where the claims are not sufficiently founded.

Answer 4: In order to improve shortcoming mentioned by the Reviewer, we added in table 1 the reference to the work [49] performed at the PHELIX. Comparison in terms of flux/sr is, to our mind, not relevant for gamma and proton driven nuclear reactions with isotropic neutron sources.

In the case of pitcher-catcher targets [46, 47, 49] discussed in table 1, the fraction of directed neutrons is presented.

The current status of our results on gamma-ray production is discussed in Answer 1.

R2 (5): Finally, I have some concerns about the author's projection for nucleosynthesis application, which they have focused mainly in this paper as a near-term possibility. There has been a significant mix-up between the neutron flux, average neutron flux, whether fast or thermal flux... while comparing with the conventional sources. Furthermore, the r- and s- process requires keV neutrons, where the (p,n) and (gamma,n) sources are of MeV energies. Hence, we will lose 3-4 orders in neutron flux while moderating the fast neutrons. The fast neutrons are indeed of sub-ns or ps duration. However, the moderation will make the neutron bursts of microsecond duration, i.e. we again lose three orders in brightness. Hence I would be cautious making strong claims of meeting the required flux based on a crude scaling/estimation.

Answer 5: We understand the concerns of the reviewer and allow ourselves to respond to them below.

Neutron energies of interest for astrophysics are between 1 keV and several hundreds of keV [13, 29, 30]. Hill et al. [30] have demonstrated theoretically the capability of laser-driven neutron sources to explore neutron capture cascades and the production of neutron-rich isotopes in the nuclear astrophysical research. Here, besides others multi-neutron capture processes are investigated starting from ^{197}Au . This isotope describes a waiting point within the r-process of heavy element nucleosynthesis. We have measured via proton-induced neutrons for the case of foams combined with thin metal foil targets a higher reaction yield of isotopes produced in the keV neutron energy range (also for the production of $^{198\text{m}}\text{Au}$) compared to the conventional known case using foils only. As described in the revised Version starting from L 435 and L 582, the spectral feature of the initial protons interacting with the high-Z sample material, allows to produce a higher neutron fluence below 1 MeV neutron energy, which is interesting for astrophysical applications. Therefore, the neutron fluxes can be conserved, because no moderation is needed.

Also, in the paper by Chen et al. [29], application of upcoming multi-PW class laser facilities at ELI-NP is discussed to trigger 3 times neutron capture processes starting from ^{96}Zr . The proton induced neutron spectrum discussed for this purpose has a Maxwellian distribution with temperature of 1 MeV peaked at 0.5-1 MeV neutron energy, very similar to what is measured at PHELIX. It should be noted that our concept was done already at moderate relativistic laser intensities with pulse energy of 20 J without special laser condition, which are needed for conventional laser-proton or -deuteron based neutron production schemes producing multi-MeV mean neutron energies.

Response to reviewer #3

General:

- Discussion into the prior NCD foam work already studied in the field, while this paper adds significant detail and discussion on the emission from these targets it is worth acknowledging the prior work in the field, and discussing their relevance herein.

Answer: Beside the experiment described in this work, only two experimental campaigns have been performed to demonstrate the DLA in sub-mm long NCD plasma at moderate relativistic laser intensity [52, 53]. Here, NCD plasma was produced by a ns ASE [52] or using a well-defined ns-pulse [53] to trigger a super-sonic ionization wave in sub-mm thick low density foams.

The most known simulations and experiments deal with ultra-high laser intensities and multi-layered targets with μm -thin foams [69-71].

[52] Willingale L. et al.. The unexpected role of evolving longitudinal electric fields in generating energetic electrons in relativistically transparent plasmas. *New Journal of Physics* 20, 093024 (2018).

[53] Rosmej, O. N. et al. Interaction of relativistically intense laser pulses with long-scale near critical plasmas for optimization of laser based sources of MeV electrons and gamma-rays. *New Journal of Physics* 21, 043044 (2019).

To this point, see text from L224.

- While important to highlight the activation lines and actual measurement of the germanium signal the data is currently presented unclearly. For example Figure 5 shows first the yield from the activation measurements and then separates the fluence as a function of energy, what matters from a scientific perspective is the

second half of the graph. My suggestion is to simplify these figures to only show the scientific aspect rather than “showing the working”, else the reader needs to first look at Fig 5a) then compare to Fig 3c) to understand what point is trying to be made. The same is true in Fig 7.

Answer: We thank Reviewer for this advice.

In the revised version of our manuscript, we have summarized the gamma spectra of the germanium detector measurements for two different laser-target setups and angular scenarios in a new Fig.7, section Methods C. Also, in a new Fig. 8, a more detailed presentation of the reaction cross sections for the identified nuclear reaction channels is shown together with the experimental observed reaction yields. In addition, in figure 8, reaction yields for two different scenarios reaching record values are exemplary selected and compared with GEANT4 simulations.

These new figures are implemented in the method section, where important information to the activation measurements is given. As the result, we have a new structure of the section in the chapter III (Results and Discussion) where the old figures 3 and 5 are removed.

Abstract:

- I think a key result is the conversion efficiency for laser-neutrons this should be included in the findings in the abstract alongside the absolute numbers

Answer: We agree and added the conversion efficiencies for the neutron as well as the MeV-photon production using the foam target systems.

Introduction:

- Conversion to $N/cm^2 s$ requires a further discussion on the temporal profile of the neutrons in this process, you mention how for the S and R process it must be less than a second and while the neutron pulse is certainly shorter than this when normalizing to $N/cm^2 s$ it matters as to what pulse length you are using.

Answer: As written from L88, the neutron capture time, where nuclear astrophysical relevant multi capture processes occur, is determined by the product of the peak-flux of neutrons and the capture cross section of the isotope of interest. This capture time have to be shorter than the decay time of the generated isotope after neutron capture. The pulse duration of the laser-driven neutron pulses is from several ps to ns. Important value is the neutron fluence (N/cm^2) within the neutrons pulse duration. In the mentioned example in the introduction (from L84) concerning multi neutron capture processes on stable ^{96}Zr isotope, it requires a neutron peak-flux of $10^{24} N/cm^2 s$ to reach a capture time of about 1 s for the production of $\{97-102\}Zr$ isotopes in multi capture processes, means the capture time is shorter than the half-life of the heavy Zr isotopes.

- L102, “planed” is misspelled

Answer: We corrected to “planned”

- L124, I think counter rather than “contra”

Answer: We corrected to “counter-propagating”

- L138, remove very. I dislike the emotive language and think it would be better to state that these conditions enable QED studies.

Answer: We corrected starting from L141: "...enable for the investigation..."

- L159, what is multi 10th?

Answer: We corrected the sentence from L 165:

The realization of a laser driven spallation neutron source needs several tens of MeV proton energies. State-of-the-art research on these neutron production concepts are presented in [46, 48, 49].

Experiment:

- L205, I think you should state that the different intensity regimes are achieved by different focussing conditions. It's stated in the methods but it is important here for context.

Answer: We added after L250: These different intensity regimes were achieved by different focusing systems [Rosmej, PPCF2020].

- L210, was the plasma density measured at all?

Answer: From L259: "Measurements of the plasma density inside the ring holder after ionization by a super-sonic wave is a big challenge. At the same time, PIC-simulations performed for a step-like density profile with n_{cr} and $0.5 n_{cr}$ [40, 41] and for a partially ramped density profile in order to account for plasma expansion toward the main laser pulse [43] showed a very similar overall behavior of the energy and angular distributions of super-ponderomotive electrons."

Pugachev L P *et al* 2016 Acceleration of electrons under the action of petawatt-class laser pulses onto foam targets *Nuclear Instruments and Methods in Physics Research A* **829** 88–93

Pugachev L P and Andreev N E 2019 Characterization of accelerated electrons generated in foams under the action of petawatt lasers *J. Phys.: Conf. Ser.* **1147** 012080

Rosmej O N *et al* 2019 Interaction of relativistically intense laser pulses with long-scale near critical plasmas for optimization of laser based sources of MeV electrons and gamma-rays *New J. Phys.* **21** 043044

- Fig1, sentence structure in the caption needs work.

Answer: We corrected the text for description of Fig.1:

Top view of the diagnostic set-ups used for irradiation of aerogel-foams at 10^{19} W/cm² (20 J in focal spot) laser intensity and conventional foils at ultra-relativistic intensity of 10^{21} W/cm² (40 J in focal spot). An imaging plate behind 6 mm thick steel cylinder was used to measure an angular distribution of electrons with energies >7.5 MeV. Nuclear activation plates consisting of different materials were placed at the front of cylinder at 5° and 15° to the laser axis. Three 0.99 T magnet spectrometer measured the energy distribution of accelerated electrons at 0° , 15° and 45° to the laser axis.

Results and Discussion:

- L246, throughout I think target is clearer than radiator for the converter foil.

Answer:

We let “target” for foam-case and for foam-foil-case and use instead of radiator the word “converter”

Also, we introduce a shorter convention for the designation of the four principal laser-target setups, as explained after L284.

- L253-256, This description is verbose, in line with my general comments above, I think some of the findings in the paper would be better presented as figures with clear results. For instance for this passage a plot of angle versus the two measured components in both target cases would show the variation much better than the written description.

Answer:

From L284, we introduced the convention for the designations of the four principal laser-target setups used in the experiment. This should help to make the discussion more transparent. In addition, in Fig.2, information to the observation angles was added.

- L262-264, this process is stochastic by nature and one would expect greater variation from this. Adding a caveat to the effect of “the mean free path is ..” would help here.

Answer: From L304:

The difference in the spectra between setups 1a and 1b is caused by propagation of electrons through 1mm thick Au-converter in the 1b-case. This thickness corresponds to the mean free path of electrons with $E \leq 8$ MeV.

- Fig 2, why is the thin target for 10^{21} Ti rather than Au, it is hard to do the direct comparison you are after with such a difference in Z and Density.

Answer: We agree with the Referee at this point. Unfortunately, no data were obtained in the case of irradiation of Au foils with ultra-relativistic laser intensities. Additional results on acceleration of electrons upon irradiation of Ti and Au-foils stacked with foams are shown in Fig.7 of PPCF 2020 Rosmej et al.

- Fig 2, the caption is ordered wrong with (b) last.

Answer: We corrected the order in the caption of fig. 2.

- Fig 2, in general I think it wise to smooth the raw electron spectra plots to avoid the very busy structure we see here. The core information you wish to present here is the increase between the conditions this can be better expressed by varying the parameter on the bottom (intensity, or angle of measurement) against the temperature or flux.

Answer: In new Fig.2, smoothed electron spectra are presented.

- L290, How are you defining maximum energy? In both Fig 2c and d there are dots above 40 MeV for each color,

Answer: The maximum measured electron energy is defined at the double differential electron number slightly below 10^9 , where a signal is still beyond the noise level.

- Fig 3, while this data is clearly important to how the measurement was done what the reader cares about is the cross-sections in Figure 3c and what the measured signal tells you about the photon/neutron spectra not the raw counts from the germanium. It makes sense to include this as part of the methods description but not the core of the paper.

Answer: We removed old Fig. 3 and added a new Fig. 7 in the section Methods, where we present the gamma spectra of the activation samples at 5° and 15° angle position of the four different laser-target scenarios. Also, a new Fig. 8 is added in the method section, which presents the reaction cross sections and the obtained nuclear reaction yields for the different cases.

- L335, given the nature of the paper being a comparison between DLA with foams and the higher intensity solid target interaction you should compare the emission cone between these two cases not the two foam shots. Do you have the emission cone data for the solid target as well?

Answer: The angular distribution of electrons during the interaction of the ultra-relativistic laser pulse with a foil was measured using three 0.99 T-spectrometers set at 0° , 15° , and 45° degrees to laser axis (new Fig.2). The spectra measured at 0° and 15° overlap each other over a wide range of energies, while the spectra measured at 45° show a slightly lower temperature and half the maximum energy. Unfortunately, no cylinder diagnostic was implemented for shots at $10^{21}\text{W}/\text{cm}^2$.

In the revision, it is discussed from L330.

- Fig 4 (+surrounding discussion), how did you inject the electron beam into the target in GEANT? As this will influence the presented result.

Answer: In the Monte Carlo simulations, the electron energy and angular distributions measured in experiments were used as input parameters. The electron number was estimated from published calibration curves of Fuji MS and TR imaging plates (Bonnet T, Comet M, Denis-Petit D *et al* 2013 *Rev. Sci. Instr.* **84** 103510, Tanaka K A, Yabuuchi T, Sato T *et al* 2005 *Rev. Sci. Instr.* **76** 013507) and geometry of magnetic spectrometer and experimental set up. The effective electron temperature was determined from the slopes of the spectra, assuming a Maxwellian-like distribution. The solid angle of an electron beam propagating in vacuum was taken equal to 0.160 sr ($\frac{1}{2} = 13^\circ$) as it was measured in the experiment.

- Fig 5, This should be reduced to just panels C and D, I don't think the upper panels add anything as the reader must jump back to the cross-sectional data to determine what these lines really mean. What is important here is the flux/temperature presented in the second half. It would also be worth including the uncertainty in energy associated with the crosssections

Answer: The reaction yields are presented in the new Fig. 8 a,b in the section Methods. We added new Fig. 3 and Fig. 6 presenting the bremsstrahlung fluences and the neutron numbers.

- L552-555, it is unclear as to if this is an incredibly directional beam with <5 msr divergence or the peak was measured over that angular range.

Answer: This is a solid angle of activation sample

- L568, “unfortunately” and “scaling”

corrected

- L570, can the proton beam divergence be estimated from other work in the field? Again these foams are not unique and have been used to drive proton emission is there a comparison that can be made to aid this discussion?

Answer: We added further experimental analysis on the proton beam angular distribution (from L 445, section “Accelerated protons”) and discuss it together with 3D PIC simulation results.

- L576, emotive language

Answer: We changed to “strong enhancement”.

- Table 1, this is hard to read with the multiple line entries for the row labels, this should be formatted differently to make it simpler. Either stretching it across the whole page or adding in lines.

Answer: The presentation of Table 1 is improved and additional data are added.

- Fig 7a, again irrelevant as Fig 7c distils the important information for the reader.

Answer: Due to a large amount of information presented in our work, we decided to remove this section and to use the results together with more detailed simulation study in the frame work of a separate paper. In the revision, the optimization is shortly mentioned from L742:

GEANT4 Monte Carlo simulations were performed to optimize the neutron production for PHELIX parameters (10^{19} W/cm², $E_{FWHM} = 20$ J, setup 1b). After placing the Au converter direct to the foam rear-side and increasing converter thickness up to 5-7mm, a record neutron fluence of 2×10^{11} cm⁻² and a flux of 10^{22} cm⁻² s⁻¹ estimated for ~20 ps neutron pulse duration was achieved.

- Fig 7b, I like this schematic, the description needs to be expanded but it could stand with D as its own figure.

See previous item.

Conclusions

- L710, the conversion efficiency I read as the main highlight from Table 1 this should be added into the abstract.

Answer: We added the conversion efficiency into the abstract

Methods

- L768, what caused the spot to be so elliptical? And is the energy in the wings of the spot

detrimental to the plasma channel in anyway?

Answer: The laser focal spot is initially elliptical. Wings contain a significant portion of the laser pulse energy, but they are much wider than the main spot and therefore less intense. Since the ponderomotive force that creates the plasma channel is determined by the transverse gradient of the relativistic laser intensity, the main plasma channel is produced by the energy in the focal spot.

- L775, don't use about, state the uncertainty.

Answer: We corrected to 38 ± 2 J.

- L785, was the density measured at all? Is the assumption of fully ionised foam valid for the conditions?

Answer: The electron density was not measured (see our answer to the question L210).

Answer: Yes, the assumption of fully ionized foam is valid.

1D two-temperature simulations with the code RADIANT [G. Verguniva et al, *Plasma Physics Reports*, 2013, Vol. 39, No. 9, pp. 755–762] that account for radiation transport in plasma were performed by interaction of 5×10^{13} W/cm² nanosecond laser pulse with CHO. It was shown, that by the direct laser irradiation of 2 mg/cc foams, the electron temperature reaches 600 eV. This is more than enough for ionization of K-shell electrons in Carbon and Oxygen. On the other hand, the optical field ionization by the relativistic laser pulse is the next process that leads to fully ionized CHO-plasma.

- L789, titanium. Was the thin foil with the foam target also Ti or was it Au.

From L812, we corrected: As foil targets, 10 μ m thick pure metallic foils (Au, Ti) and 1 mm thick pure Au targets were used.

- L800, "height"

Corrected.

- L806, delete "massive".

done

- L835, it is important here (as mentioned with Figure 5) it is important to discuss the associated errors/uncertainty with this method as you are then fitting to it to find a temperature.

Answer: From L862 (section Methods C.), we added discussion of the main uncertainties in the determination of the reaction yield and the deconvolution of the initial photon or particle spectrum. The uncertainties are presented as the error bars in the corresponding diagrams.

REVIEWER COMMENTS

Reviewer #1 (Remarks to the Author):

The current version has been modified and improved. Particularly, the analysis on products of nuclear reactions induced by intense gamma-ray beam and neutron source were presented more clearly. However, there are still a few of points waiting for response.

- 1) Lines 240-242, 341, the first sentence of Fig. 2 caption, 767, 875, a few of grammar mistakes or typos should be removed. In addition, the caption of Fig. 2 is not well presented.
- 2) Line 262, PIC-simulations seems to be the first occurrence, and should not be abbreviated.
- 3) Dashed lines in Fig. 3 are thermal exponential on the experimental data. Besides data fitted with the red dashed line, other data are presented with no more than three data points. It is confusing to perform the exponential fitting and then to obtain effective temperatures for these bremsstrahlung radiations.
- 4) Lines 407-415, it is necessary to clarify more clearly which kind of physical model / process is used for prediction of photonuclear reaction yield.
- 5) Lines 429-434, how does the author(s) obtain the following three values, 55%, 30%, and 1.7×10^{11} ? The detailed calculations should be given.
- 6) Fig. 5 shows the results of proton accelerated. But the caption of (b) presents electron angular distribution.
- 7) Lines 461-467, we suppose the author(s) used characteristic gamma-ray lines reduced from proton-driven nuclear reactions to identify these reaction products (as well as their reaction yields) and then to calculate the proton yield at the proton energy resulting in maximum of the cross sections. For $^{197}\text{Au}(p,3n)^{195}\text{mHg}$, the proton energy should be 28 MeV, instead of 14 MeV. The author(s) need check the data (or legend) shown in Fig. 5(a), which is not consistent with the presentation of main text. The gamma spectra obtained with setup 1a for different sequence of samples (Cr is at front or back) is necessarily displayed. Such gamma spectra would have the characteristic gamma lines for ^{197}mHg , ^{195}mHg and ^{52}Mn .
- 8) Fig. 6, what is the expression of Maxwellian-like energy distribution used here? It is very interesting to see the distributions shown in Fig. 6. The referee is wondering what the reaction channels that contribute the data points obtained at ~ 2.5 MeV are.
- 9) Line 775, "The simulation studies for target setup improvement" has not been presented in the current version. It should be deleted.
- 10) It is strange that the peak of 40K disappears in the background spectrum in the case of "50, foam + 10 μm Au foil" (see Fig. 7). In Fig. 7, four subfigures should be sequenced; similar to the two lower subfigures, a logarithmic scale is suggested for the two upper subfigures. Which kind of laser intensity is used to get the Fig. 7?

Reviewer #2 (Remarks to the Author):

The authors have addressed some of my comments adequately and have amended the manuscript accordingly. The revised version is more coherent and reads better.

However, I would still prefer the paper to focus more on physics rather than technical aspects. Although the article is technically detailed, the gamma and neutron measurements are based on only two activation stacks, modelling and estimations (as the authors replied, I presume low statistics, or even single-shot measurements, heavily dependent on MC fitting), which, in my opinion, does not reflect a robust measurement to adhere, and hence inappropriate for a high impact factor publication as Nat. Comm. The gamma and neutron yields/conversion are not significantly different compared to other mechanisms (for instance, [65] for gamma and [49] for neutron), albeit shown to be produced at lower intensities which make the paper interesting!

High gamma and neutron yields stem from the hot-electron population, and similar NCD interaction has been studied previously, also by the authors, showing significant boosts in electron temp. Here the authors show an unexpectedly high electron temp at 10^{19} W/cm², which in my opinion should be the main highlight of this work (gamma and neutrons are just the secondary sources governed by the cross-sections – I suppose the yields can even be improved by choosing better converters). Hence, if the high electron temp is not due to some interesting new physics (as the authors did not indicate in their reply), the paper can be considered simply a natural extension of the previous works [53,54] and, in my opinion, would not merit publication in Nat. Comm.

Reviewer #3 (Remarks to the Author):

The authors have made significant progress to both the clarity and the presentation of the paper. My original comments were mainly to address the presentation of the various activation measurements the authors used during this study, which they have now done. This paper is a thorough characterisation of neutron and gamma emission from a DLA interaction within NCD plasmas, demonstrating high conversion in comparison to higher intensity “conventional” targets. I think the work within this paper is well suited to Nature Comms, however the authors may wish to consider the following minor changes to strengthen the narrative of the paper:

- As described at the end of the experiment description there are two laser conditions and two target conditions, it might be useful to compare them like for like. Figure 2 for example might benefit from the electron spectra in a) being compared against c) and d) directly.
- Aspect ratio of figure 3 is strange to make room for the legend, if it was split into two columns it might look better.
- Figure 5a) could be cut to 10^{-4} without losing anything, c) could also be limited to 0.05 on the y axis.
- Figure 8 the plotting style changes between the panels, and the legend in a) is directly over some of the lines.

Answer to Reviewer Comments

Reviewer #1:

The current version has been modified and improved. Particularly, the analysis on products of nuclear reactions induced by intense gamma-ray beam and neutron source were presented more clearly. However, there are still a few of points waiting for response.

1) Lines 240-242, 341, the first sentence of Fig. 2 caption, 767, 875, a few of grammar mistakes or typos should be removed. In addition, the caption of Fig. 2 is not well presented.

Answer:

done

240-242 The target consisted of a several micrometers thin NCD plasma slab (0.5 ncr) and over-dense foil.

341 A weak angular dependence of the electron spectra indicates a rather divergent electron beam.

767 separation...

875 The photon or particle spectrum was deconvolved by

New caption Fig.2:

Electron spectra. Electron spectra registered at three different angles (0° , 15° , 45°) to the laser axis:

(a) Spectra measured in interaction of a laser pulse of $\sim 10^{19}$ W/cm² intensity and ~ 20 J in the FWHM of the focal spot with aerogel target stacked together with a gold foil of $10 \mu\text{m}$ thickness (setup 1a, green) and with 1 mm thick gold converter (setup 1b, red).

(b) Result of the cylinder stack diagnostic for the angular distribution of super-ponderomotive electrons with energies >7.5 MeV measured in a single shot onto aerogel+thin foil (left) and aerogel+mm-thick converter (right).

(c) Spectra measured in interaction of a laser pulse of $\sim 10^{21}$ W/cm² and 40 J in FWHM of the focal spot with $10 \mu\text{m}$ titanium foil and (d) with 1 mm-thick gold converter.

2) Line 262, PIC-simulations seems to be the first occurrence, and should not be abbreviated.

Answer:

done

3) Dashed lines in Fig. 3 are thermal exponential on the experimental data. Besides data fitted with the red dashed line, other data are presented with no more than three data points. It is confusing to perform the exponential fitting and then to obtain effective temperatures for these bremsstrahlung radiations.

Answer:

There was a mistake. We corrected Fig 3 and presented not less than 4 experimental points for every laser shot. No further results described in the manuscript were affected.

4) Lines 407-415, it is necessary to clarify more clearly which kind of physical model / process is used for prediction of photonuclear reaction yield.

Answer:

From new line 427 we add information of the physics process used in the simulation.

5) Lines 429-434, how does the author(s) obtain the following three values, 55%, 30%, and 1.7×10^{11} ? The detailed calculations should be given.

Answer:

We revised the explanation given before on the base of GEANT4 simulations.

The revised text from L460 is as following:

The beam of DLA-electrons passes a 1 mm Au-converter and produces the MeV-bremsstrahlung radiation with 0.11 sr divergence angle. A high fraction of the relativistic electrons escapes the converter (Fig. 2, 1b) with 0.27 sr divergence, propagate 23 cm in vacuum and produce additional gamma-radiation in the activation samples. According to GEANT4 simulations, 65% of the total gamma-fluence originates in the converter and 35% in the activation stack. Taking in to account 1.2×10^{12} ph/sr produced in two different ways described above, we obtain the total photon number with $E > 7.5$ MeV:

$$1.2 \times 10^{12} \text{ ph/sr.} \times (0.65 \times 0.11 \text{ sr} + 0.35 \times 0.27 \text{ sr}) = 2 \times 10^{11}$$

6) Fig. 5 shows the results of proton accelerated. But the caption of (b) presents electron angular distribution.

Answer:

corrected:

“...b) IP signal produced by electrons with $E > 7.5$ MeV directed along the laser axis together with the positions of activated samples for detection of (p,xn)-reactions and expected direction of TNSA-accelerated proton beam (foil-target normal was $+10^\circ$ to laser direction). ...”

7) Lines 461-467, we suppose the author(s) used characteristic gamma-ray lines reduced from proton-driven nuclear reactions to identify these reaction products (as well as their reaction yields) and then to calculate the proton yield at the proton energy resulting in maximum of the cross sections. For $^{197}\text{Au}(p,3n)^{195}\text{mHg}$, the proton energy should be 28 MeV, instead of 14 MeV. The author(s) need check the data (or legend) shown in Fig. 5(a), which is not consistent with the presentation of main text. The gamma spectra obtained with setup 1a for different sequence of samples (Cr is at front or back) is necessarily displayed. Such gamma spectra would have the characteristic gamma lines for ^{197}mHg , ^{195}mHg and ^{52}Mn .

Answer:

-In the text, we corrected the value of the proton energy to 28 MeV.

-The text from new line 500 and Fig. 5a are corrected.

-In Fig 7., a further gamma spectrum concerning the measurements from activation stacks with different sequences are added.

8) Fig. 6, what is the expression of Maxwellian-like energy distribution used here? It is very interesting to see the distributions shown in Fig. 6. The referee is wondering what the reaction channels that contribute the data points obtained at ~ 2.5 MeV are.

Answer:

-For more explanation of the used Maxwellian-like energy distribution, we added further text from new line 622 and from new line 645:

“The total neutron numbers were obtained by fitting the measured neutron spectra with a Maxwellian-like distribution function $N(E/(\pi E_{\text{mean}}))^{1/2} \exp(-E/E_{\text{mean}})$ using the known E_{mean} , where E is the neutron kinetic energy.”

“The neutron spectrum in the case of 1a can be explained by a combined Maxwellian-like distribution with an additional exponential part describing the high energy tail of the neutron spectrum [77-79]. Here, the process of precompound excitation in proton induced nuclear reactions is taken into account, where the step after the nuclear reaction leads to the probability to emit a nucleon (neutron) immediately before the compound nucleus state [80-82]. Such probability increases with the proton energy. Therefore, the flat slope of the proton spectrum obtained in the setup case 1a, favors the probability for precompound excitations. Neutrons emitted from the precompound state have the most energy but very few in number compared to the statistical evaporated neutrons from compound state, which means a small contribution to the neutron energy distribution [78, 79].”

-The reaction channel which contributes to the data points at 2.5 MeV is the fast neutron scattering reaction on ¹¹⁵In leading to the ^{115m}In exciting state.

9) Line 775, “The simulation studies for target setup improvement” has not been presented in the current version. It should be deleted.

Answer:

done

10) It is strange that the peak of ⁴⁰K disappears in the background spectrum in the case of “5o, foam + 10 μm Au foil” (see Fig. 7). In Fig. 7, four subfigures should be sequenced; similar to the two lower subfigures, a logarithmic scale is suggested for the two upper subfigures. Which kind of laser intensity is used to get the Fig. 7?

Answer:

The background line of ⁴⁰K appears in both, the 5° as well as the 15° case of foam+10μm Au foil. For explanation, the background level of the detector used in the 5° case was much lower. Therefore, only in the 15° case the ⁴⁰K appears stronger. This is described in the inset text of Fig. 7.

- We changed to logarithmic scale of all presented gamma spectra in Fig. 7 and show inset spectra for the case “1 mm Au converter”, which shows more details.

- All results with application of foams were obtained at $\sim 10^{19}$ W/cm²

Reviewer #2:

1. The authors have addressed some of my comments adequately and have amended the manuscript accordingly. The revised version is more coherent and reads better.

However, I would still prefer the paper to focus more on physics rather than technical aspects.

Answer: The physics of the DLA process is well described in our and other works cited in [54, 55, 56, 57, 58, 62, 63].

In the focus of this work is **an application of high current DLA electrons** for gamma and neutron generation and experimental demonstration of an ultra-high efficiency of the laser-driven sources. The position of the Reviewer is not clear to us, as numerous papers on the

application of LWFA and SM-LWFA mechanisms have been published on the NATURE platform. Why not for DLA?

2. Although the article is technically detailed, the gamma and neutron measurements are based on only two activation stacks, modelling and estimations (as the authors replied, I presume low statistics, or even single-shot measurements, heavily dependent on MC fitting), which, in my opinion, does not reflect a robust measurement to adhere, and hence inappropriate for a high impact factor publication as Nat. Comm.

Answer: Even if measurements are made, the complexity of the physical processes occurring during the laser plasma interaction cannot be fully understood without various types of simulations (PIC, MC, etc.). The MC simulation used describes very good experimentally obtained yields of nuclear reactions for all activated materials and provides an excellent tool for reliable interpretation of the results obtained. In addition, the nuclear activation method allows with high precision a selective measurement of multi-MeV photon and particle characteristics, especially in such a high-flux of mixed photon and particle scenarios obtained in high energy short laser pulse interactions with matter. The method is well established in laser-plasma experiments as well as standard in many high-energy laser laboratories.

3. The gamma and neutron yields/conversion are not significantly different compared to other mechanisms (for instance, [65] for gamma and [49] for neutron), **albeit shown to be produced at lower intensities which make the paper interesting!**

Answer: In our opinion, it is incorrect to put on the same level the theoretical work [65] on the scheme of obtaining MeV gamma radiation at ultrahigh intensities of 10^{21} W/cm² and the experimental results realized at 10^{19} W/cm² (this article). Rather, it should be surprising that the conversion efficiency is very similar in both cases, since it grows with laser intensity.

In this context, it is better to mention the LLNL results on gamma-ray production (H. Chen, Phys. Plasmas 22, 056705 (2015) and results of the NIF campaign with compound parabolic concentrators <https://lasers.llnl.gov/news/nif-creates-matter-and-antimatter-from-light>). with 0.3% laser-to gammas conversion efficiency.

4. High gamma and neutron yields stem from the hot-electron population, and similar NCD interaction has been studied previously, also by the authors, showing significant boosts in electron temp. Here the authors show an unexpectedly high electron temp at 10^{19} W/cm², which in my opinion should be the main highlight of this work (gamma and neutrons are just the secondary sources governed by the cross-sections – I suppose the yields can even be improved by choosing better converters). Hence, if the high electron temp is not due to some interesting new physics (as the authors did not indicate in their reply), the paper can be considered simply a natural extension of the previous works [53,54] and, in my opinion, would not merit publication in Nat. Comm.

Answer:

Laser accelerated electrons are of great interest for applications since they govern processes of gamma-rays and neutron production. One can add to this list also the Betatron and THz radiation, e-e+ pair production, proton acceleration, isotopes etc.

This is why the current work is focused on **applications** of DLA-electrons to boost secondary laser sources of MeV particles and radiation and **is a significant step forward** after the results published by the authors on the experimental characterization of the high-current well- directed beams of super-ponderomotive electrons.

It was pointed out by the Reviewer that number of MeV photons and neutrons per Joule laser energy obtained at moderate relativistic laser intensity **is higher than in experiments with conventional targets irradiated at 20-50x higher laser intensities.**

Moderate relativistic laser intensity is characteristic for large kJ-class ps lasers such as ARC (NIF), PETAL (LMJ), OMEGA EP etc., focused on optimization of the laser-driven sources of MeV particles and radiation for probing of high energy density matter.

In our work, we have demonstrated **for the first time** effective gamma and neutron production in experiments with foams in combination with foils irradiated with laser pulses of moderate relativistic intensity.

This approach promises to greatly increase the diagnostic potential of existing kJ laser systems used for ICF research, and is gaining even more importance with the recent success at NIF with 1.3 MJ released energy

Compound parabolic concentrator (CPC) was currently implemented on ARC, NIF to boost the effective laser intensity to one order of magnitude and to provide efficient generation of MeV electrons, gamma rays and positrons.

In our work, we present an **experimentally approved alternative to the NIF approach** to strongly enhance the coupling of the laser energy to MeV-particles and photons by using *large-scale low-density polymer foams in combination with foils or high-Z converter.*

Moreover, **foam targets can be used in combination with CPC** resulting in **more than two orders of magnitude higher effective laser intensity** that will allow **entering the GDR region** and perform experiments on laboratory astrophysics on large kJ PW-class lasers.

The proposed target set-up based on the DLA mechanism is **universal** and can be used for generation of **protons, neutrons and gammas** without changes in the laser-set-up and at low laser contrast. It demonstrates high level of reproducibility and pointing stability in contrast to laser-driven sources generated on the SM-LWFA-platform used at LLNL [e.g. J.L Shaw 2021, [www.nature.com/scientificreports](https://doi.org/10.1038/s41598-021-86523-5), <https://doi.org/10.1038/s41598-021-86523-5>].

Therefore, we strongly believe that the revised manuscript is appropriate for publication in *Nature Communications* as a multidisciplinary journal.

Reviewer #3:

The authors have made significant progress to both the clarity and the presentation of the paper. My original comments were mainly to address the presentation of the various activation measurements the authors used during this study, which they have now done. This paper is a thorough characterisation of neutron and gamma emission from a DLA interaction within NCD plasmas, demonstrating high conversion in comparison to higher intensity “conventional” targets. I think the

work within this paper is well suited to Nature Comms, however the authors may wish to consider the following minor changes to strengthen the narrative of the paper:

- As described at the end of the experiment description there are two laser conditions and two target conditions, it might be useful to compare them like for like. Figure 2 for example might benefit from the electron spectra in a) being compared against c) and d) directly.

Answer:

We changed and added text from new line 326:

“In direct shots on thin metallic foils at ultra-relativistic laser intensity of 10^{21} W/cm² (setup 2a), the electron energy distribution was approximated with $T_{\text{hot}}=9.9-2.2$ MeV (Fig. 2c) for measurements at 0° and 15° to the laser axis and the maximum of the detected electron energy reached 50 MeV, which is twice lower than in the case 1a, 1b. In shots onto 1 mm thick converter (setup 2b), the effective temperature and the maximum of the detected energies of escaping electrons are 4 MeV and 40 MeV correspondingly (Fig. 2d). The dependence of the electron energy distribution from angle to the laser axis is observed only at 45°, which indicates a rather divergent electron beam. In shots with 10^{21} W/cm² laser intensity, 100 μm entrance slit was used (Methods, Electron diagnostics). This resulted into comparable level of the electron signals and the background caused by the bremsstrahlung radiation from the Au-converter and, as a consequence, rather noisy electron spectra.

Measurements using cylinder diagnostic (see Methods) showed that the electron beam generated via DLA process in NCD plasma is well-collimated. Fig. 2b presents results of 120°- cylinder diagnostic (see methods) for setups 1a and 1b. The Imaging Plate (IP) signals are caused by electrons with $E > 7.5$ MeV, which are capable to trigger nuclear reactions in activation samples. The measurements show that in the first case, the relativistic DLA electrons with energies above 7.5 MeV propagate within a half angle of 13_ to the laser axis (0.16 sr solid angle), while in the set-up 1b within 20_ (0.38 sr) due to electrons scattering in 1 mm thick foil. From shot-to-shot deviations of the electron beam position from the laser axis was not larger than 5°. This result is in contrast to the stochastic beam pointing reported by Willingale et al. [54].”

- Aspect ratio of figure 3 is strange to make room for the legend, if it was split into two columns it might look better.

Answer:

We checked figure 3, but we don't understand what the reviewer means. We think the aspect ratio is correct.

- Figure 5a) could be cut to 10^{-4} without losing anything, c) could also be limited to 0.05 on the y axis.

Answer:

Done. Concerning 5c), we believe the color bar doesn't fit if the scale is shortened.

- Figure 8 the plotting style changes between the panels, and the legend in a) is directly over some of the lines.

Answer:

We corrected Fig. 8

REVIEWERS' COMMENTS

Reviewer #1 (Remarks to the Author):

I suggest accepting the manuscript after the following clarifications are done:

(1) In the response letter the total number of photons with energy $E > 7.5$ MeV is 2 time 10^{11} , whereas in the manuscript the total number of photons with energy $E > 10$ MeV is 2 time 10^{11} .

(2) As also pointed by Referee #3, $y_{lim}(y1, y2)$ of Fig. 3 should not be so wide to leave the space to fill out the legend. $y_{lim}(y1, y2)$ should be set from, for example, 10^8 to a few 10^{10} .

Reviewer #3 (Remarks to the Author):

The authors have tidied up the comments from my last review, overall I think the paper presents a thorough characterisation of neutron and gamma emission from a DLA interaction within NCD plasmas. The authors have made significant effort to include detailed technical information on both the experimental methodology and the complimentary monte-carlo simulations. I think the work within this paper is well suited to Nature Comms, and ready to be published now.

As a minor comment in Fig 1 the material labels should be consistent and state the full material name not a contraction, Fig 4 should have spatial axis, Fig 5c hides most of the information by trying to keep the y-axis with the same scale as the x-axis while you are trying to demonstrate the collimation of the beam you could do that with the y-axis limited to 0.1 and adding lines to indicate 5° and 10° .

Response to REVIEWERS' COMMENTS

Reviewer #1 (Remarks to the Author):

I suggest accepting the manuscript after the following clarifications are done:

(1) In the response letter the total number of photons with energy $E > 7.5$ MeV is 2×10^{11} , whereas in the manuscript the total number of photons with energy $E > 10$ MeV is 2×10^{11} .

Answer: *The number of photons with energies above 10 MeV is correct.*

(2) As also pointed by Referee #3, $y_{lim}(y_1, y_2)$ of Fig. 3 should not be so wide to leave the space to fill out the legend. $y_{lim}(y_1, y_2)$ should be set from, for example, 10^8 to a few 10^{10} .

Answer: *Done.*

Reviewer #3 (Remarks to the Author):

The authors have tidied up the comments from my last review, overall I think the paper presents a thorough characterisation of neutron and gamma emission from a DLA interaction within NCD plasmas. The authors have made significant effort to include detailed technical information on both the experimental methodology and the complimentary monte-carlo simulations. I think the work within this paper is well suited to Nature Comms, and ready to be published now.

As a minor comment in Fig 1 the material labels should be consistent and state the full material name not a contraction,

Answer: *We include the full material name as suggested from the referee #3.*

Fig 4 should have spatial axis,

Answer: *We have added axes to the 2d representations, which describe the horizontal and vertical angular range. Cartesian coordinate system is also implemented for spatial orientation.*

Fig 5c hides most of the information by trying to keep the y-axis with the same scale as the x-axis while you are trying to demonstrate the collimation of the beam you could do that with the y-axis limited to 0.1 and adding lines to indicate 5° and 10° .

Answer: *We changed the y-axis limit to 0.1 and adding lines, which demonstrate the two different angular ranges.*